

# Atmospheric Moisture Effects on Heavy Precipitation During the HyMeX IOP16 Using GPS Nudging and Dynamical Downscaling

Alberto Caldas-Alvarez[1], Samiro Khodayar[1,2]

[1]Institute of Meteorology and Climate Research (IMK-TRO), Karlsruhe Institute of Technology, Karlsruhe, P.O. Box, 76131, Germany

[2]Mediterranean Centre for Environmental Studies (CEAM), Valencia, 46980, Spain

*Correspondence to*: Alberto Caldas-Alvarez (alberto.caldas-alvarez@kit.edu@email.com)

**Abstract.** Gaining insight on the interaction between atmospheric moisture and convection is determinant to improve the model representation of heavy precipitation, a weather phenomenon that every year brings casualties and important monetary losses in the western Mediterranean region. Given the large variability of atmospheric moisture, an accurate representation of its distribution is expected to reduce the errors related to the representation of moist convective processes. In this study, we assess the sensitivity of precipitating convection and underlying mechanisms during a heavy precipitation event (HyMeX intensive observation period 16) to corrections of the atmospheric moisture spatio-temporal distribution. Sensitivity experiments are carried out by nudging a homogenised data set of GPS-derived Zenith Total Delays (GPS-ZTD) with sub-hourly frequency (10 minutes) in 7km and 2.8 km simulations with the COSMO-CLM model over the western Mediterranean region. The analysis shows that (a) large atmospheric moisture amounts (Integrated Water Vapour ~ 40 mm) precede heavy precipitation at the affected areas. This occurs 12 h before initiation over southern France and 4 h over Sardinia, north eastern Italy and Corsica (our main study area). (b) We found that the moisture is transported on the one hand, swept by a westerly large-scale front associated with an upper-level low and on the other hand evaporated from the Mediterranean Sea and north Africa. The latter moisture transport occurs in the <1 km to 4 km layer and has been identified for this event for the first time. (c) COSMO-CLM overestimated the atmospheric humidity and precipitation amount over the study region (Corsica) and this was, to a good extent, corrected by the GPS-ZTD nudging by reducing noticeably both quantities, bringing results closer to observations. (d) The two processes that exerted the largest control on precipitation were the reduction of atmospheric instability over the island (CAPE -35 %) and the drying of the lower free troposphere bringing more dry air entrainment. Besides, the 7 km simulation showed a stronger impact for large-scale dynamical lifting at the target area, given a weakening of the represented low-pressure system and the associated wind circulation. This reduced ultimately, the intensity and number of convective updrafts represented over the island. These results highlight the large impact exerted by moisture corrections on precipitating convection and the chain of related processes leading to it across scales. Additionally. The modelling experiments demonstrated the benefit of sub-hourly GPS-ZTD nudging to improve the modelling of precipitation.



## 1 Introduction

Heavy precipitating convection causes yearly serious damages and casualties in countries of the Western Mediterranean (WMed) basin especially by autumn (Llasat et al., 2010; Gilabert and Llasat, 2017). During these events daily accumulated precipitation over 150 mm is not rare and precipitation rates can reach 20 mm h$^{-1}$. These are caused mainly by convective events ranging several temporal and spatial scales, from the mesoscale down to the micro-alpha (Ducrocq et al., 2016; Funatsu

et al., 2018). Accurate representation of the convective processes interacting across-scales is crucial to support forecasters and decision makers to prevent impacts on properties and communities. The WMed is especially prone to heavy precipitating convection by autumn because of the combination of the relatively high sea surface temperature of the Mediterranean and the Atlantic, the arrival of low-pressure systems such as extra-tropical cyclones or upper-level troughs and the interaction with the Mediterranean complex orography. Former studies pointed out the synoptic situation conducive to heavy precipitation as

usually dominated by a low-pressure system, inducing a south-westerly warm and moist inflow, building sufficient instability and moisture convergence (Jansa et al., 2001; Toreti et al., 2010; Nuissier et al., 2011; Ricard et al., 2012; Xoplaki et al., 2012, Khodayar et al. 2016). These studies also demonstrate the key role of atmospheric moisture at all phases of convective development and the need of gaining knowledge regarding its interaction with convection across scales to improve the modelling of extreme phenomena (Sherwood et al., 2010; Ahrens and Samson, 2010). Given the high spatio-temporal

variability of atmospheric moisture, a deficient representation of its distribution (Steinke et al., 2015; Girolamo et al., 2016) has been pointed out as a source of uncertainty in current predictions (Chazette et al., 2015; Khodayar et al., 2016). That is why; there is growing interest in developing forecast systems that assimilate humidity observations with sub-hourly frequency (Guerova et al., 2016). Given the novelty of such assimilation frequencies and the multiple methodologies applied, new insights are needed on their impact on simulated atmospheric conditions leading to heavy precipitation.

Determinant for the development of precipitating convection in the WMed are the vast moisture amounts associated with the heaviest precipitation events, which may originate from remote or local sources (Ricard et al., 2012; Krichak et al., 2014; Khodayar et al., 2016). Depending on the synoptic conditions, the Mediterranean Sea can account for > 50 % of the transported moisture (Duffourg and Ducrocq, 2011). This is the case when an anticyclonic flow dominates the 3 to 4 days preceding heavy precipitation. Remote sources such as the Atlantic and the tropics also supply the needed moisture, especially for the heaviest

precipitation events (Pinto et al., 2013; Winschall et al., 2014), whose transport is brought via tropical plumes (Chazette et al., 2015b) or extra-tropical cyclones (Xoplaki et al., 2012). Finally, north Africa has been also identified as a source by Duffourg and Ducrocq (2011). Despite being less common, this source provides larger amounts of specific humidity when low pressure over the Iberian Peninsula induces cyclonic conditions. Regions prone to convective development are correlated with areas of moisture convergence where building up of conditional instability takes place (Ricard et al., 2012; Khodayar et al., 2016).

The variability and stratification of atmospheric moisture determines when, where and how intense, convection can be. Several studies (Duffourg and Ducrocq, 2011; Ricard et al., 2012; Khodayar et al., 2016, Maranan et al., 2019) have highlighted the moistening of the lower troposphere as a necessary factor for precipitating convection to develop with specific humidity values



up to 10 g kg$^{-1}$ below 850 hPa. A moist low-level, increases the Convective Available Potential (CAPE-ML) energy of the lifted parcel. A second factor, crucial for convection intensity is the moisture at the Lower Free Troposphere (LFT). This is

the moisture transport that occurs above the Planetary Boundary Layer (PBL) where the influence of the surface roughness can be considered negligible. Recent observational studies (Virman et al., 2018; Schiro and Neelin, 2019) concluded that the probability of intense convection increases rapidly with increasing LFT humidity, especially over land. In this regard, a more humid LFT prevents larger dry air entrainment from happening. Khodayar et al. (2018) quantified relative humidity to be > 75 % at 700 hPa in the location of all convective systems during a Heavy Precipitation Event (HPE). In addition to an increased

probability of transition to deep moist convection, a more humid LFT enhances convection intensity (Zhuang et al., 2018). Whether this sensitivity of heavy precipitation to LFT moisture variations is well represented by current atmospheric models has been investigated by past sensitivity modelling studies using fine model resolutions, from Δx~ 500 m to Δx~7 km (Keil et al., 2008; Honda and Kawano, 2015; Lee et al., 2018). They demonstrated that convection enhancing/weakening happened when increasing/diminishing moisture at the LFT in the simulations. These studies performed moisture sensitivity experiments

modifying the water vapour distribution by adding or subtracting a prescribed water vapour amount at chosen heights. It is thus, of particular interest to investigate the aforementioned issues by performing corrections toward observations instead of using idealized experiments.

Given the correlation between the location of moisture convergence and precipitating convection, the complex Mediterranean orography plays a decisive role in setting areas prone to heavy precipitation. The high mountain ridges constrain the moisture

transports in the basin favouring moisture gathering at the mountain foothills, the coasts and the valleys. Moreover, the elevated terrain provides dynamic lifting to the convergent moist air masses triggering convection. The mountain slopes bring the low-level air masses to the level where they become buoyantly unstable. Therefore, the Alps (Italy, Switzerland, and Austria), the Massif Central (France) and Corsica (France) are focal regions for precipitating convection events (Ducrocq et al., 2014). The case of Corsica is especially characteristic given the complex distribution of valleys and ridges, which induces diurnal

variations in the mountain atmospheric boundary layer coming from processes related to the terrain (Adler et al., 2015). This induces spatial inhomogeneities in the water vapour distribution that are crucial to determine the timing and location of deep convection (Adler et al., 2015) The linear composition of the highest peaks in the northwest to southeast direction render the island prone to heavy precipitation. Corsica is one of the main study regions of this paper where we assess relevant aspects of the moisture and convection interactions for a HPE coinciding the Intensive Observation Period (IOP) 16 of the Hydrological

Cycle in the Mediterranean Experiment (HyMeX; Ducrocq et al. 2014) field campaign in autumn 2012.

In relation to the problem of accurately representing heavy precipitation, the combination of recent advances in remote sensing techniques for atmospheric moisture measuring and the growing computational power has enabled the achievement of relevant improvements through data assimilation (Wulfmeyer et al., 2015). A well-established method to assimilate data is the Nudging scheme (Schraff and Hess, 2012), where the main advantages are its simplicity (Guerova et al., 2016) and that it has shown

good results especially in analysing humidity fields as compared to other schemes (Schraff et al., 2016). Nudging can be used to assimilate Global Positioning System (GPS) measurements that provide information on the total column atmospheric





moisture. The demonstrated benefits of using GPS measurements are that it is an all-weather product (as opposite to other remote sensing integrated products), its large accuracy and its high temporal resolution (Cress et al., 2012; Guerova et al., 2016). The GPS data set used for nudging in this work is provided in the framework HyMeX. This unique HyMeX GPS
product is particularly interesting given the common processing of data from more than 25 European and African networks bringing a dense coverage of the area and its temporal resolution of minutes (Bock et al., 2016). The total number of stations included in the nudging sums up to over 900 in the whole WMed and specifically over Corsica up to 20. In this sense, an open question is what the different impacts of nudging GPS data across resolution simulations are. Especially after reaching grid lengths that explicitly resolve convection (< 3 km). With this purpose, we use two horizontal different horizontal resolutions
(7 km and 2.8 km) to quantitatively asses the different impacts of correcting the atmospheric moisture distribution depending on the corresponding model configurations.

Within this framework, this work is devoted to assessing the benefit of atmospheric moisture corrections with state-of-the-art GPS-derived measurements on sub-hourly time frequencies for the modelling of heavy precipitation through realistic sensitivity experiments.

We analyse this issue first by understanding the role of local and remote atmospheric moisture contributions to the convection-related processes leading to the occurrence of the event, and second through moisture sensitivity experiments nudging GPS information. The IOP16 of the HyMeX Special Observation Period (SOP) 1 has been extensively investigated in the past by e.g. Thévenot et al. (2015), Duffourg et al. (2016) and Martinet et al. (2017). This study complements those previous publications providing a detailed analysis of the relevance and characteristics of atmospheric moisture for the same case.

The organization of the paper is as follows. Section 2 describes the model set-ups and the modelling experiments and presents the observational data sets used for model validation or nudging. Section 3 provides a description of the event including the synoptic situation, the convective evolution and the transport of moisture. Section 4 discusses the impact of the GPS nudging in precipitation, humidity and convective-related processes and Sect. 5 presents the conclusions.

## 2 Observations and Model Set-ups

### 2.1 Observations

*GPS-Zenith Total Delay*

The Zenith Total Delay (ZTD) is the "excess path length of GPS satellite emissions (in the L1 and L2-band) caused by the refractivity of the neutral atmosphere" (Businger et al., 1996). The refractivity definition for the neutral atmosphere depends on the partial pressures of water vapour and dry air and on the temperature as introduced in ground papers of GPS meteorology
(Bevis et al., 1994). The ZTD is proportional to the Integrated Water Vapour (IWV) in the zenith direction. The ZTD is given in length units and the delay in the Zenith direction is usually preferred given it shows the shortest delays. It is obtained from the slanted path delays by means of mapping functions, $Z_m(\theta)$ dependent on the curvature of the Earth and the elevation angle (Duan et al., 1996). The dataset used for the sensitivity experiments, is provided by the LAboratoire de Recherche en Géodésie



(LAREG) and the HyMeX community and its specifications can be found in Bock et al. (2016). It merges data from more than
25 European and African networks, with over 900 stations, is made available in temporal resolutions up to five minutes and it
has a dense coverage of the western European countries (see Fig. 1). All networks have been commonly processed by the
GIPSY-OASIS II software to guarantee homogeneity. Data screening includes outliers, range and ambiguity checks to increase
the accuracy. The comparison against radiosonde IWV measurements has shown no significant biases during night-time and
biases in the range 0.5 – 1.4 mm during daytime (Bock et al., 2016).


*Radiosondes*

In the framework of the HyMeX SOP1 MétéoFrance provides the operational soundings containing more than 30 atmospheric
parameters, including temperature, dew point temperature, wind speed, geopotential height, air pressure, wind direction and
wind speed. In average, they contain ca. 30 levels between the surface and the 300 hPa level with about one measurement
every 250 m. In addition to the operational soundings, supplementary soundings were launched during the HyMeX IOPs.
Hence, the temporal resolution of the soundings lies between 12 h and 6 h. In total, 10 stations are used among which 3
(Gibraltar, Mallorca and Dar El Beïda) are used for process-understanding and 7 over the Italian Peninsula, Croatia, Corsica
and Sardinia are used for validation of the specific humidity and IWV simulations. We perform the validation of the model
data obtaining the nearest grid points to the location of the radiosondes. No height correction is applied for this purpose since
the difference in height between the neighbouring grid points and the height of the radiosonde stations does not exceed 30 m
in any case. The data is accessible at http://mistrals.sedoo.fr/?editDatsId=595&datsId=595&project_name=HyMeX.

*Rain gauges*

Météo-France and the HyMeX program provide the HyMeX domain precipitation amount (Nuret, 2013; SEDOO OMP. doi:
10.6096/mistrals-hymex.904) data set with hourly accumulated precipitation measured by rain gauges. Over 5000 stations are
deployed over western Mediterranean land parts with about 30 stations placed over the island of Corsica. The version used
(V4) enjoys the newest quality control and checks for double stations. The data set spans the Sep-2012 to Mar-2013 period.

*The NOAA CPC Morphing Precipitation (CMORPH)*

CMORPH makes available precipitation measurements in a rectangular grid merging satellite microwave observations. These
are combined using the Morphing technique (Joyce et al., 2004, https://doi.org/10.5065/D60R9MF6), that uses motion vectors,
derived from infrared data to transport the microwave information to spots where no microwave data were available. It has a
broad coverage (60°S - 60°N), and its spatial and temporal resolution at the equator is of 8 km and 30 minutes. The Climate
Prediction Center (CPC) of the National Weather Service (NWS) in the USA provides the data and it spans the period 1998 to
2015. CMORPH has shown a good detection skill in validation studies (Bitew and Gebremichael, 2011; Habib et al., 2012)
and high correlation rates with sub-daily rain gauge data (Sapiano and Arkin, 2009).



*Global Land Evaporation Amsterdam Model (GLEAM)*

GLEAM provides daily accumulated terrestrial evaporation (evapotranspiration) in a 0.25° x 0.25° grid, spanning the period
2003-2017. GLEAM computes the total evaporation over land from the sum of bare-soil evaporation (Eb), transpiration (Et), open-water evaporation (Ew), Snow sublimation (Es), snow sublimation (Es) and interception loss (Ei), as described in Martens et al. (2017) and Miralles et al., (2011). Four interconnected modules dealing with the rainfall interception, soil stress, soil state and the evaporation calculation, compute the aforementioned contributions. The four modules are forced by gridded global data which, in the version used in this work (v3b), is obtained mostly from remote sensing products, such as the Clouds
and the Earth's Radiant Energy System (CERES) for radiation, the Tropical Rainfall Measurement Mission (TRMM) for precipitation, the Atmospheric Infrared Sounder (AIRS) for air temperature or European Space Agency Climate Change Initiative Soil Moisture (ESA CCI SM) for soil moisture. GLEAM version v3b has shown an average correlation with in-situ measurements of 0.78. In the validation, only 2 out of 63 stations showed differences with a level of significance of 10 % (Martens et al., 2017).


*The Hybrid Single-Particle Lagrangian Integrated Trajectory Model (HYSPLIT)*

The HYbrid Single-Particle Lagrangian Integrated Trajectory (HYSPLIT) computes air parcels, dispersion and chemical transformations (Stein et al., 2015; Rolph et al., 2017). In this paper, we use HYSPLIT to compute backward trajectories of moisture sources. The HYSPLIT model uses a hybrid approach combining lagrangian trajectories with the Eulerian
methodology, using a fixed three-dimensional grid as a frame of reference (Stein et al., 2015). The free access internet-based platform READY (https://www.ready.noaa.gov/index.php) offers HYSPLIT trajectories calculation using eight different atmospheric model analyses of meteorological data. In this work, we use the half-degree archive of the National Centers for Environmental Prediction (NCEP) Global Data Assimilation System (GDAS) that spans the period 2007 to present and has a global coverage. The dataset is accessible in https://www.ready.noaa.gov/HYSPLIT_traj.php, last accessed 18-July-2019.

**2.2 The COSMO-CLM Model and the GPS-ZTD Nudging Sensitivity Experiments**

**2.2.1 Consortium for Small-scale Modelling (COSMO) in Climate Mode**

The COSMO model is based on the fully compressible, nonhydrostatic, hydro-thermodynamical equations of the atmosphere. Where the latter is considered as a multicomponent continuum constituted by, liquid water, dry air, water vapour and solid water in the form of cloud droplets ice crystals, raindrops, rimed aggregates, hail and graupel (Schättler et al., 2016). The
COSMO version used in this study is the 5.00 and the model is used in climate configuration (COSMO-CLM). This implies that the slow-changing variables (ozone concentration, aerosol concentration and canopy variables) evolve in time. This brings a more realistic representation for seasonal simulations such as the ones presented in this work. The dynamic solver is a third order Runge-Kutta split-explicit scheme following Wicker and Skamarock (2002). It uses an Arakawa-C/Lorenz grid with scalars defined at the centre of the grid box and the normal velocity components defined on the corresponding box faces. The





grid is rotated, and the height coordinate shows a Gal-Chen terrain-following grid stretching. The model uses a sponge layer

with Rayleigh damping at the top boundary and three grid point lines for adaptation at the lateral boundaries. The boundary

and initial states of the atmospheric prognostic variables are obtained by coarser resolution forcing models in a one-way nesting

approach. The soil state and the surface-atmosphere interactions are simulated through the TERRA-ML model (Doms et al.,

2011). TERRA-ML has eight soil layers and is responsible for issuing the temperature and humidity conditions at the ground

and considers the processes of evaporation, runoff, snow storage and interception storage. COSMO-CLM in the used

configurations for the resolutions of this work (for a 7 km and a 2.8 km grid resolution), parameterizes the turbulent diffusion

using a 1D diagnostic closure for the turbulent kinetic energy (Doms et al., 2011). The grid-scale clouds and precipitation are

parameterized using a bulk scheme including several hydrometeor types (Doms et al., 2011). The radiation is parameterized

following the formulation after Ritter and Geleyn (1992). In the case of grid spaces larger than 3 km sub-grid deep moist

convection is parameterized using a mass-flux, low-level scheme with the equilibrium closure based on moisture convergence

(Tiedtke, 1989). Shallow convection is parameterized using an adaptation of the Tiedtke scheme in the simulations using a 7

km and a 2.8 km grid.

*The Nudging Scheme*

Following Schraff and Hess (2012), "nudging or Newtonian relaxation consists of relaxing the model's prognostic variables

towards prescribed values within a predetermined time window". The extent of the relaxation depends on the difference

between the observed and modelled variable ($[\varphi_k^{obs} - \varphi(x_k, t)]$), a weighting factor considering the measurement location

and its representativeness ($W_k(x, t)$) and a coefficient that modulates the impact of the analysis increments ($G_\varphi$). This term is

added to the result provided by the dynamics and numerics ($F(\varphi, x, t)$) see Eq. (1.a). For an observation type, the weighting

factor considers the spatial and temporal distance to the observed value. The spreading in the horizontal direction $w_{xy}$ is

performed following a second-order autoregressive function of the distance between the location of the observation and the

target point ($\Delta r/s$ ; see Eq. 1.b). The vertical interpolation of the observed data is performed assuming a Gaussian decay in

height differences.

$$\frac{\partial}{\partial t} \varphi(\boldsymbol{x}, t) = F(\boldsymbol{\varphi}, \boldsymbol{x}, t) + G_\varphi \cdot \sum_{k_{obs}} W_k(\boldsymbol{x}, t) \cdot [\varphi_k^{obs} - \varphi(\boldsymbol{x}_k, t)] \qquad (1.a)$$

$$w_{xy} = (1 + \Delta r/s) \cdot e^{-\Delta r/s} \qquad (1.b)$$

Regarding temporal weighting, for hourly or even more frequent data measured from a stationary platform, the data are

temporally interpolated linearly to the model time. The observations are assigned to a grid point in the spatio-temporal space

and the body of the report is evaluated. This is the step where gross error and consistency checks, quality control and

redundancy checks dismiss suspicious observations.

In the case of GPS-ZTD observations, these are converted to Integrated Water Vapour (IWV) following Bevis et al. (1994).

Making use of run-time pressure and temperature values interpolated to the station location, COSMO-CLM computes IWV

from the issued ZTD (Schraff and Hess, 2012). The observations are assigned to a grid point in the model space, provided the





altitude difference of the GPS station and model surface lays within the range -150 - 600 m to allow for extrapolation and interpolation, respectively and are converted to a specific humidity profile ($q_{v_{obs}}$). This is needed given IWV is not a model

prognostic variable. The profile is constructed ($q_{v_{mod}}$) by means of an iterative process that scales the observed IWV ($IWV_{obs}$) with the modelled IWV ($IWV_{mod}$) until a sufficiently low error is reached, see Eq. (2). The first constructed profile ($q_{v_{mod}}$) for the iterative process, is the modelled specific humidity profile. Hence, the profile used for nudging depends on the vertical humidity distribution simulated by the model at the beginning of the nudging time-window.

$$q_{v_{obs}} = q_{v_{mod}} \cdot \frac{IWV_{obs}}{IWV_{mod}} \qquad (2)$$

**2.2.2 The GPS-ZTD Nudging Sensitivity Experiments**

The Nudging scheme is used to assimilate GPS-ZTD data to assess the sensitivity of heavy precipitating convection to corrections of the spatio-temporal distribution of atmospheric moisture. The methodology is described as follows, we perform reference runs, hereafter referred to as CTRL, of the period 1-Sep 0000 UTC to 20-Nov 0000 UTC using two different horizontal resolutions (7 km and 2.8 km). Subsequently, we simulate the same period, keeping the same settings but nudging

GPS-ZTD data continuously every 10 minutes. These runs are called NDG-7 and NDG-2.8. The 7 km runs (CTRL-7 and NDG-7) have been forced by European Centre for Medium-Range Weather Forecasts (ECMWF) analyses. The 2.8 km runs (CTRL-2.8 and NDG-2.8) are forced by the CTRL-7 simulation in a one-way nesting strategy. The simulation domains are contained in Fig. 1. Within the 80-day period of simulation, there are several events, which are largely affected by the GPS-ZTD nudging. IOP16 is one of them, which is especially interesting given the large precipitation reductions (-20 %) and the

important role of the local orographic and instability factors in triggering and maintaining convection rather than the large-scale upper level forcing. Under a weak synoptic forcing, the impact of the GPS-ZTD is larger given the strongest correction of the lower to middle tropospheric humidity. We validate the model output against in-situ humidity measurements quantifying the Mean Absolute Error (MAE) and Mean Bias (MB) and the Agreement Index (AI) as described in González-Zamora et al. (2015). The precipitation fields are validated against rain gauges and the evapotranspiration over land using spatial averages

of the GLEAM product.

To investigate the impact of moisture variations on convection-related processes, such as atmospheric latent and potential instability conditions, several convective related indices are examined. The CAPE-ML, providing information about the latent instability, is obtained through the mean layer parcel method, as described in (Leuenberger et al., 2010). KO-index is obtained as the differences in θe between several levels of the atmosphere up to 500 hPa (Andersson et al., 1989) hence it bears

information on potential instability and how the upper-levels introduce atmospheric instability. Finally, the moisture flux is obtained by multiplying specific humidity and the horizontal wind following Ricard et al. (2012) but computed at each pressure level.



## 3 Atmospheric Moisture transport and Precipitating Convection during IOP16

IOP16 was an HPE that took place on the 26-Oct 2012 where several convective systems affected regions in the north-WMed.
Radar measurements showed precipitation intensities over 100 mm in the course of 6 h, brought by large Mesoscale Convective Systems (MCSs) located close to the Spanish and French coasts displacing inland in the course of the day. Rain gauge stations measured totals up to 200 mm d$^{-1}$ over southern France, 150 mm d$^{-1}$ south of the western Alps and 250 mm d$^{-1}$ at the gulf of Genoa (250 mm d$^{-1}$). Over Corsica, most precipitation occurred over the western side of the island for twelve hours, with maximum precipitation intensities reaching 25 mm h$^{-1}$. Once the active systems reached land, the cold pools and the mesoscale
moist south-westerly flow, as well as the local orographic lifting further promoted convection (Duffourg et al., 2016).

### 3.1 Synoptic conditions

The synoptic situation during the IOP16 was characterized by a cut-off low displacing westerly from the Iberian Peninsula toward southern France between 25-Oct 1200 UTC and 27-Oct 0000 UTC (Thévenot et al., 2015). The upper levels showed an associated diffluent flow with a south-westerly to southerly circulation at the low levels over the western part of the basin.
Such a synoptic situation is prototypical for HPEs in the WMed (Jansa et al., 2001; Duffourg and Ducrocq, 2011). Over the Thyrrenean Sea on the morning of 26-Oct, the low-level induced convergence to the south of France. Figure 2 shows the geopotential height of the 500 hPa level (FI500), the Pressure at the Mean Sea Level (PMSL) and the spatial distribution of the Equivalent Potential Temperature ($\theta_e$) at 850 hPa at three hours of the event as represented by CTRL-7. Figure 2.a shows the situation on the 26-Oct 0600 UTC where values of $\theta_e$ larger than 320 K take place. This moist and energetic southerly
flow agrees with past publications referring values of the wet bulb temperature ($\theta_w$) over 16 °C (Duffourg et al., 2016; Martinet et al., 2017). Six hours later (Fig. 2.b) the cut-off Low progressed toward the border of France and Spain further deepening. On the 26-Oct 1800 UTC, the high values of $\theta_e$ (> 320 K) finally reached Corsica as well as extensive parts of the Thyrrenean Sea (see Fig. 2.c). Northerly cold winds terminated the event in the early morning of 27-Oct.

### 3.2 Convective evolution over Corsica

In the early morning of 26-Oct convection triggered and organized into a V-shape MCS close to the north-eastern coast of Spain. This MCS was named MCS0 by Thévenot et al. (2015) and hereafter we adopt the same nomenclature. This is shown by Fig. 3.a, depicting the convective cloud tops through brightness temperature images obtained with the Spinning Enhanced Visible Infrared Imager (SEVIRI) in a composite image at 0000 UTC, 0730 UTC and 1400 UTC. Northward, in the proximity of the Gulf of Lions, new convective cells triggered forming an MCS (MCS1) at 0500 UTC that would split into two defined
MCSs (MCS1a and MCS1b). The two split MCSs moved toward southern France and eastward respectively in the course of 6 h (Duffourg et al., 2016). At the southern French region, the first MCS induced precipitation accumulations over 70 mm. South of the Alps, the second split MCS brought values over 90 mm (see Fig. 3.b, showing the accumulated precipitation between 26-Oct 1300 UTC and 27-Oct 1500 UTC for the whole WMed). Another MCS developed at the Liguria-Tuscany



region in north-western Italy around 0730 UTC named MCS2 after Thévenot et al. (2015), not shown here. This area shows
the highest precipitation rates of the event with over 245 mm in 24h (Duffourg et al., 2016). High convective cloud tops are
also observed over the mid Mediterranean west of Corsica at 0730 UTC as shown by the brightness temperature (Fig. 3.a).
This shows that convection is already happening offshore before the cells arrive at the island. At 1400 UTC over the island,
the offshore convection is reinforced by orographic lifting of the moist low-level air masses.

Over Corsica, which is our study region, precipitation total values reach maximum accumulations between 75 mm and 100
mm, over the windward side of the mountains and over the mountain crests, between 50 mm and 75 mm (see Fig. 3.b). At the
lee side of the mountain, the accumulated precipitation reaches 30 mm. The first convective cells occur over the island between
1300 UTC - 1500 UTC on the 26-Oct, forced by orographic lifting precipitating with intensities up to 11.5 mm h$^{-1}$ over the
windward side of the mountains (not shown). Between 26-Oct 1900 UTC and 27-Oct 0100 UTC, offshore-size convective
systems arrive at the island (see Fig. 3.a). This stage has the largest precipitation intensities of the event (up to 16 mm h$^{-1}$, not
shown) with precipitation falling mostly over the western part of the island, transitioning from the north at 2100 UTC to the
south at 2300 UTC.

### 3.3 Atmospheric moisture transport

The transport of moisture feeding the convective systems along Corsica and at southern France and north-eastern Italy arises
from the action of the upper-level pressure low through two mechanisms. First, the associated front swept atmospheric moisture
from the Atlantic to the Mediterranean in the course of 36 h. Second, from intense evaporation over the Mediterranean and
north-Africa between 25-Oct 1800 UTC and 26-Oct 1200 UTC transported by the southwesterly flow.

In this section, we use observations from radiosondes, the Evapotranspiration product GLEAM (see Sect. 2.1), CMORPH
precipitation estimates, backward trajectories and the COSMO-CLM CTRL-7 simulation for understanding of the transport
and distribution of moisture towards the WMed region. We use the CTRL-7 simulation given the good agreement against
radiosonde measurements from the HyMeX database (discussed later in Sect. 4.2).

Figure 4 shows the CTRL-7 representation of IWV between 25-Oct 1200 UTC and 26-Oct 1200 UTC in the western
Mediterranean at three hours with the simulated wind fields at 850 hPa. At 25-Oct 1200 UTC (Fig. 4.a), the front associated
with the pressure low west of the Iberian Peninsula swept large IWV amounts, up to about 40 mm over the strait of Gibraltar
and along the southern Portuguese coast. Local areas at the Gulf of Lions (southern France) also show values as large as 40
mm at about 1200 UTC before precipitation initiation. At 26-Oct 0000 UTC, the Atlantic moisture is already located over the
Algerian coast and at the Gulf of Lions (see Fig. 4.b). As introduced in Thevenot et al. (2015) and Martinet et al. (2015), the
large moisture amount present at the Gulf of Lions originates partly from the Mediterranean Sea due to evaporation along the
Spanish eastern coast. Along the Algerian coast, these high moisture amounts at 26-Oct 0000 UTC were a combination of
moisture swept by the low-pressure system from the Atlantic and moisture evaporated from north African land. At the hour of
precipitation initiation over Corsica (26-Oct 1200 UTC), vast IWV amounts surrounded the western and southern coasts of the
island (see Fig. 4.c). These large IWV values (~ 40 mm) surrounded the island about 4 h prior to precipitation initiation. We





have additionally validated these moisture transports against the Moderate Resolution Imaging Spectroradiometer (MODIS) daily product but we do not show it here given the presence of extensive cloud cover during the period.

To better asses the timing and vertical stratification of the moisture transport the time-height cross-sections of Equivalent

Potential Temperature ($\theta_e$), specific humidity and horizontal winds measured by radiosondes between 24-Oct and 27-Oct 0000 UTC at Gibraltar (Spain), Dar el Beïda (Algeria) and Mallorca (Spain) are shown in Fig. 5. The cross-section of Gibraltar (Fig. 5.a) shows high specific humidity values throughout the complete atmospheric column (3 g kg$^{-1}$ at 500 hPa and 11 g kg$^{-1}$ at 950 hPa) between 24-Oct 1800 UTC and 25-Oct 0600 UTC, in agreement with the simulated IWV in Fig. 4.a. Further east, over Dar el Beïda (Algeria, see Fig. 5.b), high specific humidity and $\theta_e$ values are present between 500 hPa and 800 hPa on

the 25-Oct 0000 UTC. The high humidity (6 g kg$^{-1}$) and $\theta_e$ (325 K) at this time come from the arrival of the moisture swept from the Atlantic. At lower levels (800 hPa to 1000 hPa), high humidity and $\theta_e$ only happens at the station 24 h later, at 26-Oct 0000 UTC. This is because below 800 hPa, the mesoscale circulation has a strong south-to-south-easterly component (Duffourg et al., 2016; Martinet et al., 2017) which delays the arrival of the moist air masses. This can also be seen in the wind direction over Dar el Beïda on the 25-Oct 1200 UTC below 850 hPa, where a south-easterly circulation takes place. Over

Mallorca, a similar vertical distribution can be observed (see Fig. 5.c.). On the 25-Oct 0000 UTC, specific humidity values as high as 6 g kg$^{-1}$ exist between 500 hPa and 800 hPa. Twelve hours later, the high $\theta_e$ and specific humidity can be found in the layer 700 hPa to 800 hPa due to the delayed arrival of moisture at the low-levels. Finally, at 26-Oct 0000 UTC high $\theta_e$ and specific humidity is located at the marine boundary layer over Mallorca.

A relevant source of moisture feeding convection in our area of study originates in Algeria, from the Atlas Mountains and the

Kasdir region (both regions are contained within the black box NA in Fig.1). This moisture anomaly results from an intense episode of evapotranspiration over North African land between 20-Oct and 27-Oct as a result of heavy precipitation on the 19-Oct and 20-Oct. The resulting atmospheric moisture is transported by the south-westerly flow within the layer 500 m to 4 km during the evening of the 25 Oct merging with the Atlantic moisture swept by the large-scale front.

Figures 6.a and 6.b show the spatial distribution of hourly evapotranspiration simulated by CTRL-7 on the 25-Oct at 1200

UTC and 1800 UTC and the vertical updrafts of wind speed larger than 0.25 m s$^{-1}$ (Fig 6.a). Hourly evaporation rates of 0.3 mm h$^{-1}$ took place at the southern part of the NA box and of 0.2 mm h$^{-1}$ over the Algerian Atlas (northern part of the NA black box). The moisture gathers in the PBL for several days until the first convective updrafts take place over the area (25-Oct 1800 UTC, see Fig. 6.b). The radiosondes over Dar el Beïda (Fig. 5.b) show the accumulation of moisture in the lower-atmosphere (about 10 g kg$^{-1}$ close to 1000 hPa on the 25-Oct 1200 UTC). The first convective activity over NA starts at about 25-Oct 1800

UTC (see Fig. 6.b). Vertical transport of humidity is promoted by convection and continues during the evening of 25-Oct (see Fig. 6.b). At 26-Oct 0600 UTC, intense convective activity takes place over the Mediterranean Sea, between the coast of Algeria and the island (Corsica and Sardinia). This further promotes the moisture uptake form the Sea aided by the intensified drag of the low-level convergence associated with convection.

Figure 6.c represents the evapotranspiration anomaly over North Africa showing the temporal evolution of spatially averaged

precipitation measured by CMORPH and simulated by CTRL-7 and of evapotranspiration as shown by GLEAM and CTRL-



7. CMORPH shows spatially averaged daily precipitation up to 15 mm d$^{-1}$ at North Africa on the 19-Oct and 20-Oct. Precipitation during these days coincided the IOP15 of HyMeX where a large band of south to north precipitation impacted regions of Africa, eastern Spain and southern France. CTRL-7 represents this precipitation but overestimates the amount. Suite to the precipitation event, daily evapotranspiration over the area reached spatial averages of 1.4 mm as shown by GLEAM,

lasting for seven days, well above the mean evapotranspiration values during this season (0.5 mm). Albeit differences in the magnitude of evaporation, COSMO-CLM well captures this period of anomalous evapotranspiration.

Figure 6.d shows the 24h backward trajectories obtained with the HYSPLIT model, using reanalysis data from the GDAS model (see Sect. 2.1) with a resolution of 0.5°. We initialize the trajectories in a 3° x 2.5° box, every 0.5° at a height of 4000 m a.s.l., representative of a height of convective activity. The trajectories start on the 26-Oct at 1800 UTC. The trajectories

confirm the moisture transport from northern Africa, as they are located over the intense evapotranspiration area "NA" black box on the 25-Oct 1800 UTC. This is the hour when convective activity occurs over "NA" lifting the humid air masses (see Fig. 6.b). Two sets of trajectories are distinguishable. The first set (ellipse A in Fig. 7) shows faster trajectories, whose starting point is located over Morocco on the 25-Oct 1800 UTC and whose transport occurs at higher levels (between 2000 and 3000 m a.s.l). The second set (ellipse B in Fig. 7) shows trajectories which are slower, that originate over northern Algeria at a

height < 1000 m, representative of the well mixed diurnal layer (Garratt, 1994) and rise to 4000 m over the island.

## 4 Nudging Effects on Convection

The present section examines the sensitivity of the precipitation field and the underlying convection-related processes responsible for the IOP16 event, to realistic atmospheric moisture corrections through GPS-ZTD nudging.

### 4.1 Precipitation

The COSMO-CLM simulations were able to represent the event over the island on both the 7 km and the 2.8 km configurations at the right time, albeit they overestimated the amount. Indeed, the maximum accumulated precipitation simulated by CTRL-7 was between 125 mm and 200 mm and for CTRL-2.8 was between 100 mm and 125 mm, both of which are too large in comparison with the measurements (between 75 mm and 100 mm, see Fig. 3.b).

CTRL-7 performed well in representing the location of maximum precipitation; over the windward slope of the Corsican

mountains (see Fig. 7.a). The represented precipitation was mostly orographically triggered together with an offshore convective line west of the island triggered by low-level convergence (not shown). Offshore precipitation accumulations at this location brought by the convective line are between 50 mm and 75 mm. CTRL-2.8 showed a worse representation of the location of the maximum as it shifted it towards the crests of the mountain mainly and also to the lee side (see Fig. 7.c), but represented better the amount than CTRL-7. CTRL-2.8 represented more isolated precipitation structures, located over the

mountain crests.





The main differences in precipitation representation of CTRL-2.8 (Fig. 7.c) in comparison with CTRL-7 (Fig. 7.a) are the location of maximum precipitation over the crests, the missing of the offshore convective line at 26-Oct 2100 UTC and the much more localized heavy precipitation structures. The latter is a well identified effect of reaching convection permitting resolutions (Chan et al., 2012; Fosser et al., 2016).

The GPS-ZTD nudging induced for both model resolutions a decrease in the accumulated totals, bringing values closer to the observations. In the case of NDG-7, it showed maximum precipitation totals between 100 mm and 125 mm (-20 % variation), rain gauges showed precipitation between 75 mm and 100 mm (see Fig. 7.b). The location of the maximum was very similar to that of CTRL-7, over the windward side of the mountain, in good agreement with the observations. However, the convective line ahead of the island is not captured by the NDG-7 simulation because of relevant changes in the low-level mesoscale wind

circulation (not shown). These differences in wind circulation arise partly due to changes in the pressure distribution of the event, as explained in Sect. 4.4. The NDG-2.8 showed maximum accumulated precipitation in the range 75 mm to 100 mm (-25 %) over one of the mountain peaks in better agreement with observations (Fig. 7.d). The location of precipitation maxima did not change however significantly as it erroneously remained over the mountain crests.

## 4.2 Atmospheric Moisture

To assess the accuracy of model moisture outputs and the impact of nudging GPS-ZTD, independent humidity measurements from radiosonde profiling of the atmosphere are compared with the CTRL and NDG simulations. In total, we selected 55 soundings from 7 stations (blue squares within the 2.8 km simulation domain in Fig. 1), during the period 26-Oct 0000 UTC to 28-Oct 0000 UTC. The temporal resolution of the radiosondes is between 6 h and 12 h depending on the considered station. Table 1 represents the MAE, the AI and the MB of IWV for the comparison of the simulations against the 55 soundings. The

reference runs, CTRL-7 and CTRL-2.8, show MAE values of 2.7 mm, with Mean Biases of 0.61 mm and 0.38 mm and AI of to 0.88 and 0.89, respectively. The values of Mean Biases are of the same order as the RMSE values found in other publications of IWV comparison between model data and GPS observations (Bock et al., 2016). We can see that the 2.8 km slightly outperforms 7 km in representing IWV.

Nudging GPS-ZTD data, improves the scores. The MAE of IWV is 2 mm for NDG-7 and NDG-2.8 and the MB is of -0.04

mm and -0.08 mm, respectively. In this sense, both the 7 km and the 2.8 km simulations endure an improvement

Figure 8.a shows the spatially averaged temporal evolution of IWV over Corsica. The hours prior to precipitation initiation (26-Oct 1300 UTC) were characterized by an IWV pick up starting at 26-Oct 0000 UTC. All simulations show this, albeit the IWV amount over Corsica for NDG-7 and NDG-2.8 was 5 mm higher than for CTRL-7 and CTRL-2.8. This was due to represented precipitation over the island until the night of 24-Oct in the NDG runs, hence inducing a much wetter boundary

layer (not shown). By 26-Oct 1000 UTC, an intense moisture increase takes place over the island. As described in Sect. 3.3, this is the time when the Atlantic, Mediterranean and African moisture reached the island. At this time, all simulations show the same mean IWV (27 mm) which lasts for 4 hours. This is so since only after the moist air masses reach the island the GPS nudging has a noticeable impact on the simulated IWV. Given no GPS stations are located over Africa or the Mediterranean





Sea, the south-westerly flow is only weakly impacted by the GPS-ZTD nudging during the first stages of the event over

Corsica. At 26-Oct 1400 UTC, the CTRL and NDG runs start to diverge and between 26-Oct 1600 UTC and 27-Oct 0600
UTC, NDG-7 and NDG-2.8 show ca. 4 mm less than their CTRL counterparts do. This has stringent consequences for the
intensity of convection and precipitation with a vast decrease of precipitation amount, as discussed in Sect. 4.1.

The humidity reduction between 26-Oct 1600 UTC and 27-Oct 0600 UTC takes place below 500 hPa. Fig. 8.b shows how
median IWV decreases from 30 mm to 27 mm as a result of the GPS-ZTD nudging in the 7 km simulations (- 10 %) and from

30 mm to 28 mm in the 2.8 km (-7 %). At 500 hPa, a specific humidity reduction of 0.2 g kg$^{-1}$ took place for median values in
the 7 km simulation (-13 %). The decrease was weaker in the 2.8 km grid with a reduction of 0.5 g kg$^{-1}$ (33 %). At 500 hPa
the specific humidity decrease ranged between 0.5 g kg$^{-1}$ and 1 g kg$^{-1}$ for median values (-8 %) for both resolutions. At 950
hPa, the humidity reduction was larger in the 7 km (-8 %) than in the 2.8 km run (-2%).

Figure 9 represents the MAE and the MB of specific humidity profiles between 500 hPa and 950 hPa for the same set of

radiosondes. Between 600 hPa and 950 hPa, the MAE of specific humidity of CTRL-7 and CTRL-2.8 is between 0.7 g kg$^{-1}$
(600 hPa) and 1.3 g kg$^{-1}$ (925 hPa). The MB of the profile shows that this error comes from an underestimation of specific
humidity by COSMO-CLM below 650 hPa, which is largest below 900 hPa. Over 650 hPa, the simulations overestimated the
specific humidity. The GPS-ZTD nudging improves the MAE of the humidity profile between 650 hPa and 925 hPa for both
resolutions. The MAE of NDG-7 is now within the range 0.6 g kg$^{-1}$ (600 hPa) and 1.1 g kg$^{-1}$ (925 hPa) and the improvement

reaches the 950 hPa level. For NDG-2.8, the MAE is between 0.8 g kg$^{-1}$ (650 hPa) and 1.2 g kg$^{-1}$ (900 hPa) but an improvement
is only achieved down to 925 hPa. The MB is closer to zero at the same atmospheric layers (650 hPa to 900 hPa) for both
resolutions albeit showing better results for the 7 km simulation. The correction for LFT moisture is larger in the 7 km runs
than in the 2.8 km, probably due to the larger number of observations included in the nudging in this simulation because of
larger simulation domains (see Fig. 1). These values of the MAE and Mean Bias are of the same order as the validation of the

RMSE of specific humidity profiles between reanalyses data and Lidar measurements from Duffourg et al. (2016). Below 900
hPa, the GPS-ZTD nudging was not able to bring such a clear correction, especially for NDG-2.8 where the MAE and MB
showed very similar values to the CTRL-2.8 runs. The GPS-ZTD is not able to correct sufficiently the dry bias of the model
below 900 hPa because the radiosondes showed a steeper gradient of increasing humidity at the lowest levels. Both CTRL
runs show difficulties in representing this gradient and the correction induced by the GPS-ZTD nudging is not enough to moist

sufficiently the PBL during this event. Overall, COSMO-CLM shows a good performance in representing the integrated
atmospheric moisture fields and humidity over 900 hPa at both model resolutions. The 2.8 km simulation was initially more
accurate, but the nudging brings both to similar accuracy rates.

## 4.3 Instability reduction and increase of free-tropospheric mixing

The two affected processes, which exerted the largest control on precipitation reduction, were atmospheric latent instability

reduction and dry air entrainment, both investigated in this section. The drying brought by the GPS-ZTD nudging over Corsica
dried the atmosphere below 500 hPa. In this section, we will discuss how the impact was primarily reducing CAPE-ML and





additionally enhancing mixing with dry air above the PBL. The changes in these two processes start to play a role immediately after the first hour of large IWV differences i.e. after 26-Oct 1600 UTC this is so for both the 7 km and the 2.8 km simulations. Figure 10 shows the height-surface cross-sections of Equivalent Potential Temperature ($\theta_e$), specific humidity and the wind in

the vertical-horizontal direction along the direction of the mean wind (purple transect in Fig. 1) over the island at 26-Oct 1700 UTC. CTRL-7 and CTRL-2.8 show $\theta_e$ values over 322 K from the surface up to 500 hPa showing the upward transport of moist low-level air masses. After applying the GPS-ZTD nudging, NDG-7 shows reduced values of $\theta_e$ (310 K) close to the ground over the island and at 700 hPa (312 K) showing a less favourable environment for convection development. The 2.8 km simulation, for its part, showed a weak reduction of $\theta_e$ at the windward side of the mountain (316 K) as a result of the

GPS-ZTD nudging compared to the NDG-7. However, at the lee side between 600 hPa and 900 hPa, $\theta_e$ is reduced in NDG-2.8 by -8 K (compared to CTRL-2.8, 318 K), this is shown in Fig. 10.c and Fig. 10.d. The consequence for the updrafts was a change in their timing location and intensity (see Fig. 10.d). For the time shown, the drier environment in the NDG-7 and NDG-2.8 runs impedes the development of deeper updraughts.

Figure 11 shows that median CAPE-ML is reduced as a result of the GPS nudging for both resolutions from 310 J kg$^{-1}$ in

CTRL-7 to 190 J kg$^{-1}$ in NDG-7 (-39 %) and from 600 J kg$^{-1}$ in CTRL-2.8 to 410 J kg$^{-1}$ in NDG-2.8 (-32 %). Since COSMO-CLM selects the lowest 50 hPa as the mean layer to compute CAPE- ML (mixed layer), a decrease of humidity close to ground implies a relevant impact on atmospheric instability conditions. COSMO-CLM in the 2.8 km resolution represented larger latent instability than 7 km for this event. Median KO-index increased from -2.7 K, in CTRL-7 to -1.5 K in NDG-7 (+ 44 %) where lower KO-index indicates more potential for storm development under favourable large-scale conditions. The narrower

simulation domains of the 2.8 km simulations (see Sect. 2.2.2) render the impact of the GPS nudging on KO-index weaker given the inability to represent changes on the large-scale pressure distribution. The overall decrease in the median moisture flux implies a drier ground level and a drier LFT. This means that the air entrained in the convective updrafts is drier than that of the reference runs (CTRL). The median moisture flux is reduced by about 13 % in NDG-7 and about 5% in CTRL-2.8 at 700 hPa. At the PBL, the moisture flux is also reduced. The changes in moisture flux between CTRL-7 and NDG-7 are larger

than their 2.8 km counterparts. This is due to two factors, first, the changes in specific humidity are slightly weaker in the 2.8 km runs as compared to 7 km and second, the wind speed and direction in the 7 km runs are modified as a result of the GPS-ZTD nudging. For instance, at 950 hPa, extreme horizontal wind speeds are reduced by -8 % from CTRL-7 to NDG-7. This impact is not observed in the 2.8 km runs.

Overall, the humidity reduction caused by the GPS-ZTD nudging, locally over Corsica, reduced the amount of instability (as

shown by CAPE-ML and KO-index) as well as humidity at the LFT (demonstrated by the changes in specific humidity and moisture flux).

### 4.4 Impact on the low-pressure system and mesoscale winds

Besides impacting the representation of the local conditions of humidity, instability and buoyancy, the GPS-ZTD nudging affected the representation of the low-pressure system. This section shows how a large humidity reduction over the Iberian



Peninsula and France weakened the intensity of the pressure Low and its associated circulation. This brought, in turn, stringent
        modifications of the wind fields close to Corsica down to the ground and hence on dynamic lifting. This effect was observed
        in the 7 km resolution runs exclusively given the broader extent of the simulation domains. Hence, for the analysis of the
        impact of the GPS-ZTD nudging on the large-scale surface pressure distribution we focus in the 7 km resolution simulations.
        In the early morning of 26-Oct-2016, the centre of the upper-level low was located over the north-western part of the Iberian

Plateau. The GPS nudging induced moisture reductions of 7 mm in IWV at that location in the NDG-7 simulations, with very
        large reductions in the range 1-2 g kg$^{-1}$ from the ground up to 700 hPa (not shown). The progression of the pressure-low toward
        southern France was effective in twelve hours and at 1500 UTC, the PMSL was of 995hPa at the Rhône Valley (CTRL-7).
        The centre of the Low extended toward the Alps at 2300 UTC. Drying of the atmospheric column, due to the GPS-ZTD
        nudging, also took place at this region between the 25-Oct and the 28-Oct (not shown). At 2300 UTC on the 20-Oct, over the

Cévennes-Vivarais area, differences in IWV were of 3 mm between CTRL-7 and NDG-7. Figure 12 shows the differences in
        PMSL on the 26-Oct 2300 UTC between NDG-7 and CTRL-7 as well as the wind fields at 950 hPa. The GPS-ZTD nudging
        increased by 10 hPa the PMSL at the centre of the system and up to 2 hPa between Brittany (France) and the Balearic Islands
        (Spain). The impact for the cyclonic wind circulation was a veering from a south-westerly to west-south-westerly flow and a
        reduction of the wind speeds. The largest impact was observed at the 950 hPa level albeit relevant differences affecting Corsica

exist between 850 hPa and 1000 hPa. The reduction of the wind speeds is demonstrated through box-whisker plots in Fig. 11.
        This difference in wind speed does not exist in the 2.8 km runs. The consequence for convection initiation was that weaker
        low-level convergence was represented in the NDG-7. Either for offshore convergence or convergence forced by the
        orography. This hampered orographic lifting at the mountain foothills and ahead of the island reducing triggering of new cells
        and weakening the convective updrafts.

**5 Conclusions**

        Further knowledge of the pathways of moisture and convection interaction is needed (Stevens, 2005; Sherwood et al., 2010;
        Ahrens and Samson, 2010). A deeper understanding of moist processes is relevant to improve the representation of heavy
        precipitation by numerical atmospheric models in order to support the mitigation and prevention of its hazards. The presented
        work aimed at assessing the sensitivity of the precipitating convection and underlying mechanisms to realistic corrections of

the atmospheric moisture distribution. We did this by, first understanding the role of local and remote atmospheric moisture
        contributions to the occurrence of the event and second, through moisture sensitivity experiments nudging GPS-ZTD
        observations. The unique opportunity provided by the synergy of high-resolution atmospheric modelling, very frequent data
        nudging and high-resolution humidity datasets enables the study of moisture and convection interactions in a selected case
        study of heavy precipitation. With this purpose, we presented an in-depth analysis of HyMeX IOP16 with special focus on the

complex orographic region of Corsica.



The results showed novel insights on the role of remote and local moisture transports and the moisture distribution pre-conditioning heavy precipitation at the WMed and over Corsica during IOP16. These results supplement the findings by (Thévenot et al., 2015; Duffourg et al., 2016; Martinet et al., 2017), focusing in southern France and the Gulf of Lions as study region. The main findings of this study are summarized in the following

- Large atmospheric moisture amounts (IWV~ 40 mm) precondition the areas of convective activity, namely, southern France and the Gulf of Lions, Corsica and Sardinia, the middle Mediterranean and northeastern Italy, in agreement with previous investigations in the region (Khodayar et al., 2016). These very wet air masses reach southern France and the Gulf of Lions about 12 h prior to precipitation initiation and about 4h for the central Mediterranean,. A southerly to southeasterly flow at the low levels over the Mediterranean Sea delays the arrival of low-level moisture at the easterly locations.

- The transport of moisture feeding the convective cells arises from the action of the upper-level pressure low through two mechanisms. On the one hand, the associated front, sweeps atmospheric moisture from the Atlantic to the Mediterranean in the course of 36 hours. These large-scale moisture transports are delayed 24 h at the layers below 800 hPa with respect to the higher levels due to the southerly flow bringing low-level convergence south of the Gulf of Lions as referred in Martinet et al. (2017) and in Duffourg et al. (2016). This was demonstrated by the radiosondes in the region (Gibraltar in Spain, Dar el Beida in Algeria and Mallorca in Spain). Secondly, evaporated moisture over the Mediterranean and north Africa between 25-Oct 1800 UTC and 26-Oct 1200 UTC transported by the south-westerly flow towards Corsica contributed to the feeding of the convective systems. For the first time, to the authors' knowledge, this evapotranspiration anomaly over northern Africa has been identified as a source of moisture feeding HPE in the WMed during this event. Intense evapotranspiration over the Algerian Atlas and NA takes place between 21-Oct and 28-Oct as a result of intense solar radiation and precipitation impacting the region on the 19-Oct and 20-Oct. Subsequently, the intense south-westerly flow and the convective activity across the WMed Sea brings the moisture up north in the course of 24 to 30 hours as shown by backward trajectories and model output.

COSMO-CLM was able to represent the event over Corsica with a good agreement on the timing for both resolutions and an overall good performance in humidity representation. In this regard

- The reference runs using both a 7 km and a 2.8 km grid performed overall well in reproducing the moisture distribution during the event. The GPS-ZTD nudging improved the representation of IWV reducing the MAE of IWV to 2 mm. Regarding the vertical distribution of humidity, the GPS-ZTD nudging improved by 25 % the MAE from 650 hPa down to the ground in the 7 km simulation and by 8% between 650 hPa and 925 hPa in the 2.8 km resolution. The weaker improvement below 925 hPa in the 2.8 km resolution is because the model is unable to represent the abrupt humidity gradient of the lowest layers. Even if the GPS-ZTD correction is beneficial, it is not enough to overcome the humidity biases at this level. The accumulated precipitation amount was overestimated by the reference runs of both resolutions, most notably by CTRL-7 with totals between 125 mm and 200 mm at the windward side of the mountains. CTRL-2.8 also overestimated the maximum accumulated precipitation (between 100 mm and 125 mm)



this time at the mountain crests. The GPS-ZTD reduced the maximum accumulated precipitation to 100 mm in NDG-7 and to 75 mm in NDG-2.8, bringing closer values to the rain gauges observations (between 75 mm and 100 mm). The CTRL-7 and NDG-7 better captured the location of precipitation than the 2.8 km simulations. The nudging did not improve this aspect in the 2.8 km resolution.

The impact of the GPS-ZTD nudging has been assessed with the following conclusions:

- Heavy precipitation showed a large sensitivity to the moisture variations, implying a strong reduction of the maximum totals (-20 % for 7 km and -25 % for 2.8 km) arising from less intense convection and a lower number of triggered cells. This is related to the reduction of specific humidity below 500 hPa by -10 % in the 7 km and by -7 % in the 2.8 km.

- The two affected processes which exerted the largest control for precipitation reduction were the reduction of atmospheric instability over the island (-35 % CAPE-ML) and the drying of the LFT bringing more dry air entrainment into the convective updrafts (-13 % moisture flux at 700 hPa for 7 km and -5 % for 2.8 km).

- Additionally, the 7 km simulations showed an impact on the large-scale surface pressure and the associated circulation given the larger simulation domains. The GPS-ZTD nudging dried the atmospheric levels over Iberia and France, weakening the Low (~10 hPa higher for PMSL). This induced in turn a decrease in wind speed (-7 %) and a veering of the direction toward a west-south-westerly.

This study highlights the added value of adequate corrections of atmospheric moisture for modelling of convective-processes, in this case through sub-hourly GPS-ZTD nudging. The high-temporal resolution of the GPS-ZTD observations facilitate a better representation of the water vapour variability and a better regulation of the accumulated precipitation. This was shown
to be the case for HyMeX IOP16 at convection permitting and convection parameterized grid lengths. This is especially relevant since, in spite of the consensus in the scientific community that convection permitting is the future of NWP, coarser resolution simulations will still be needed as providers of forcing data and therefore reducing moisture uncertainties at these grid lengths is needed to improve the precipitation forecasts. Also noteworthy is the large sensitivity to variations of the LFT moisture showed by the model. Recent observational studies have highlighted the linkage between intense convective
precipitation and a humid free troposphere (Schiro and Neelin, 2019; Virman et al., 2018), hence the relevance of the ability to represent such sensitivity. Just as the high-temporal resolution, the dense spatial coverage and the accuracy are the clear benefits of GPS-ZTD nudging, this study also points out one of its drawbacks. Being an integrated quantity, GPS-ZTD nudging struggles to correct the vertical distribution of humidity, in this case particularly in the lower-troposphere. Lastly, this study focuses on a single case study; therefore, the results presented here should be extended to other events of the region to prove
their general applicability. In a further publication, the authors evaluate the impact of GPS-ZTD nudging on the SOP1 period of the HyMeX campaign considering the sensitivity of all IOPs in this autumn season.

## 6. Code Availability

The COSMO-CLM model is only accessible to members of the Climate Limited-area Modelling Community and access is
granted upon request. Parts of the model documentation are freely available at http://www.cosmo-
model.org/content/model/documentation/core/default.htm

## 7. Data Availability

Two further publications using these nudging simulations are on-going. Therefore, these are not yet available for the open
public. However, the data used to produce the figures showing results on the nudging simulations (Figures 2, 4, 6.a, 6.b, 6.c,
7, 8, 9, 10, 11, 12) are accessible in at https://doi.org/10.5445/IR/1000097457. The observational data used in the figures within
this manuscript are obtained from the referenced data sets and their access depends on the restrictions of the producing
institutions.

## 8. Author Contributions

SK designed and planned the experiments. ACA carried out as part of his PhD the nudging experiments under the supervision
of SK. ACA and SK analysed the results and wrote the manuscript.

## 9. Competing Interests

The authors declare that they have no conflict of interest.

## 10. Acknowledgements

We acknowledge Oliver Bock and the LAboratoire de Recherche En Géodésie (LAREG) of the French Institute of the
Geographic and Forest Information (IGN) for their providing of the GPS-ZTD data set. We also acknowledge Météo-France
and the HyMeX program for supplying the Rain gauges and radiosonde data supported by grants MISTRALS/HyMeX and
ANR-11-BS56-0005 IODA-MED. We would like to thank the German Weather Service (DWD) and the CLM-Community
for their providing of the COSMO-CLM model, and especially Ulrich Schättler and Christoph Schraff for their support in
carrying out the nudging experiments. We are also thankful to the European Centre for Medium-Range Weather Forecasts
(ECMWF) for their Integrated Forecasting System (IFS) analyses. We thank as well the teams of HYSPLIT at the NOAA Air
Resources Laboratory, CMORPH at the Climate Prediction Center (CPC) and GLEAM for their data sets. Finally, we would
like to thank the hosting institution, the Karlsruhe Institute of Technology (KIT).





## 11. Financial support

This research work has been supported by the Bundesministerium für Bildung und Forschung (BMBF; German Federal Ministry of Education and Research) project PREMIUM 01LN1319A.

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







**Table 1. Mean Absolute Error (MAE), Agreement Index (AI), and Mean Bias (MB) of IWV between the model and the observations.**
**The selected observations are all radiosondes available in the period 26-Oct 0000 UTC to 28-OCT 0000 UTC at the 7 stations within the 2.8 km simulation domain (see red box in Fig. 1). All model values have been interpolated to the location of the radiosonde station from the nearest neighbours. The difference between model and station height never exceeded 30 m.**

| Rads vs. COSMO-CLM (IWV) | MAE [mm] | AI | MB [mm] |
|---|---|---|---|
| CTRL-7 | 2.7 | 0.88 | 0.61 |
| NDG-7 | 2 | 0.93 | -0.04 |
| CTRL-2.8 | 2.7 | 0.89 | 0.38 |
| NDG-2.8 | 2 | 0.92 | -0.08 |



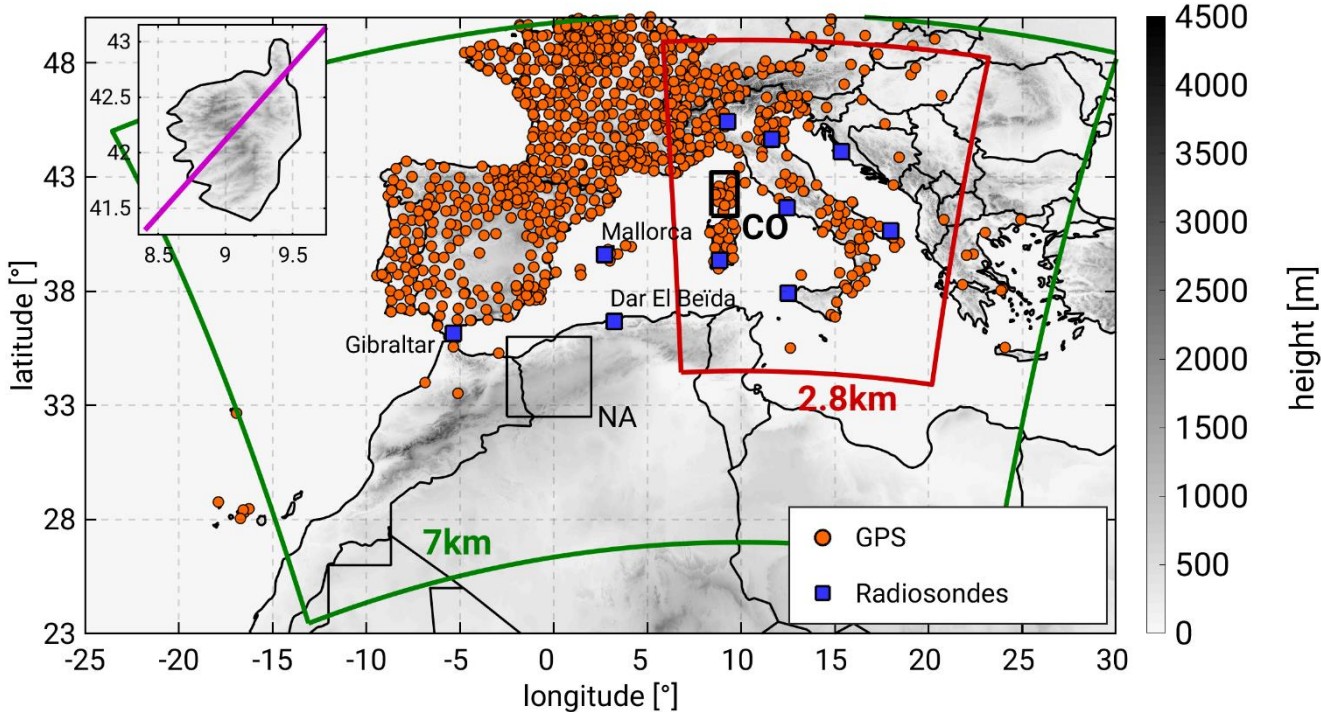

**Figure 1: Simulation domains for the CTRL-7 and NDG-7 runs (green line) and for the CTRL-2.8 and NDG-2.8 (red line). The investigation area Corsica (CO) is depicted by the thick lined black box. The box over North Africa (NA) denotes the region of intense evapotranspiration between 20-Oct and 27-Oct (see Sect. 4.1). The orange scatter points show the location of the GPS receivers used for nudging and the blue squares the location of the radiosondes used for validation and process-understanding. The upper left corner shows a detail of the orography over CO with a purple transect used for representation of the humidity vertical distribution in Fig. 5.**




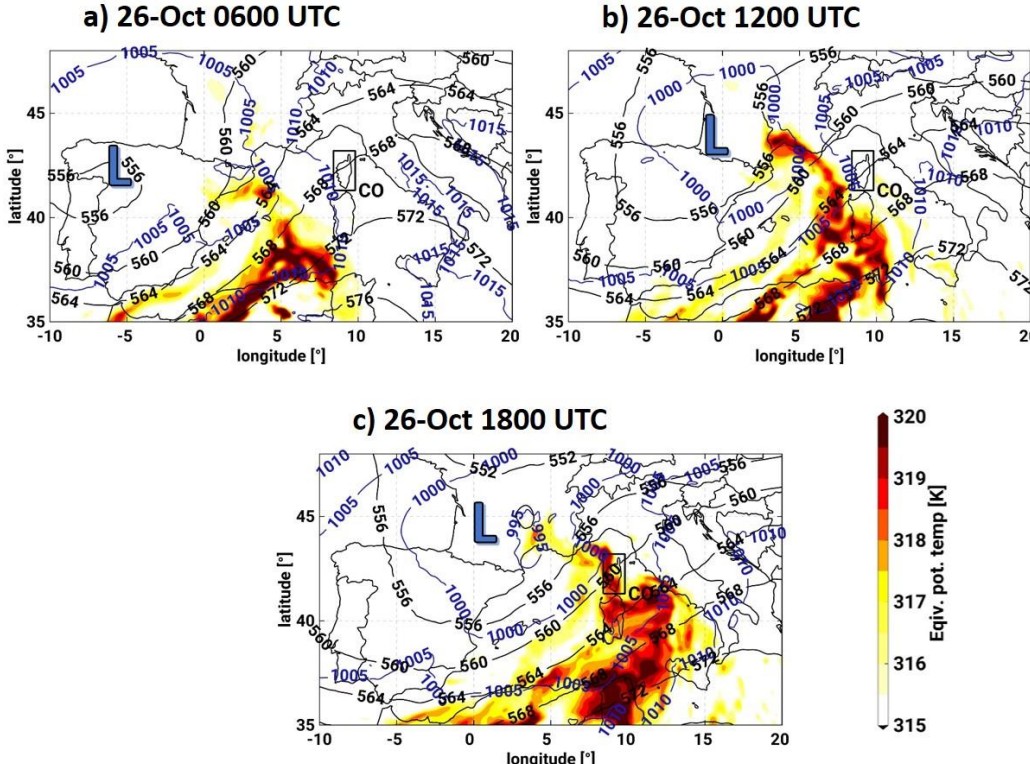


**Figure 2. CTRL-7 simulation of 500 hPa geopotential height in black isolines (gpdam), mean sea-level pressure in blue isolines (hPa) and equivalent potential temperature at 850 hPa in colour scale (K) on the 26-Oct-2012 at 0600 UTC (a), 1200 UTC (b) and 1800 UTC (c).**


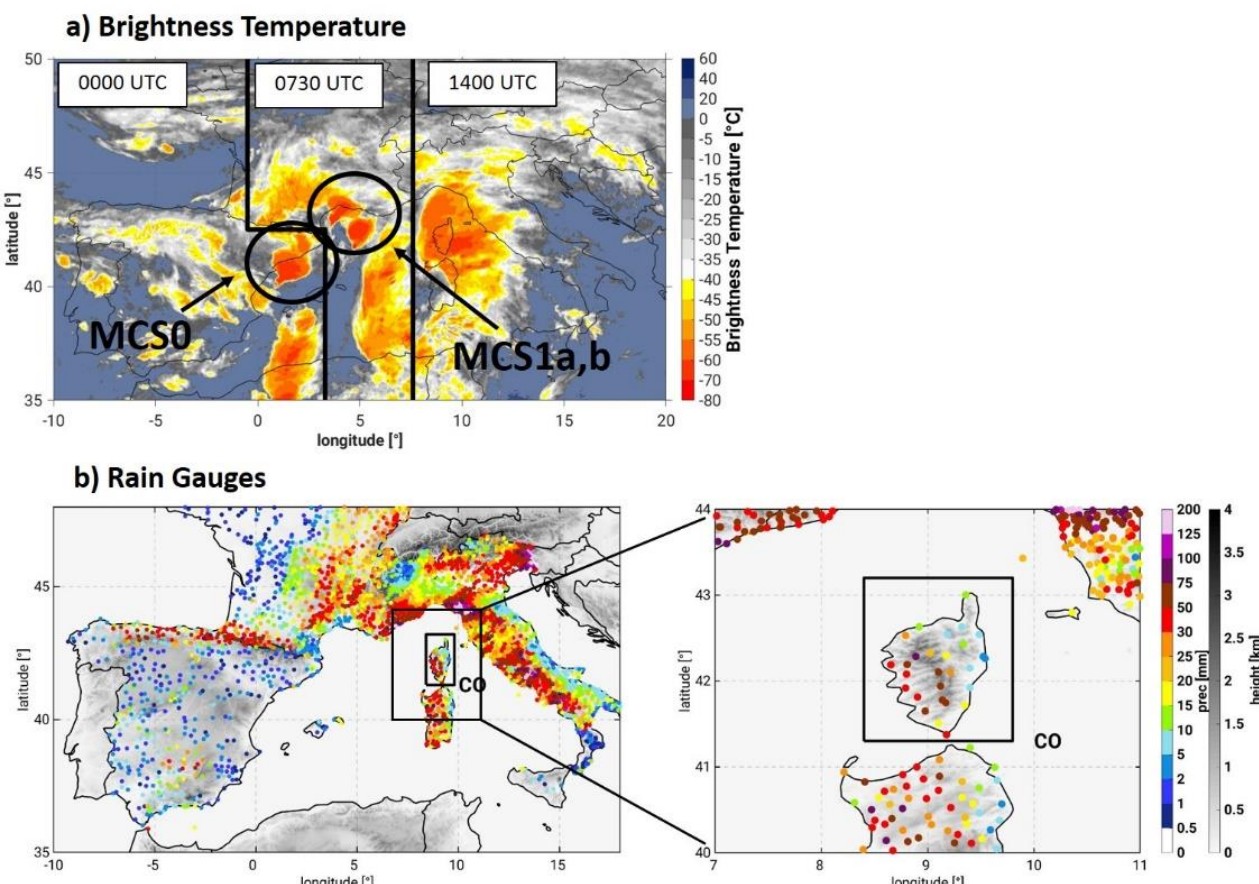

**Figure 3. Composite image of brightness temperature as measured by the SEVIRI satellite on the 26-Oct-2012, at 0000 UTC, at 0730 UTC and at 1400 UTC (a). Rain gauge accumulated precipitation over the complete Western Mediterranean (WMed) region between 26-Oct 1300 UTC and 27-Oct 1500 UTC and zoomed over the study region of Corsica (b). The accumulation period shown is the period of heavy precipitation over our main study region, the island of Corsica (black box), during this event.**
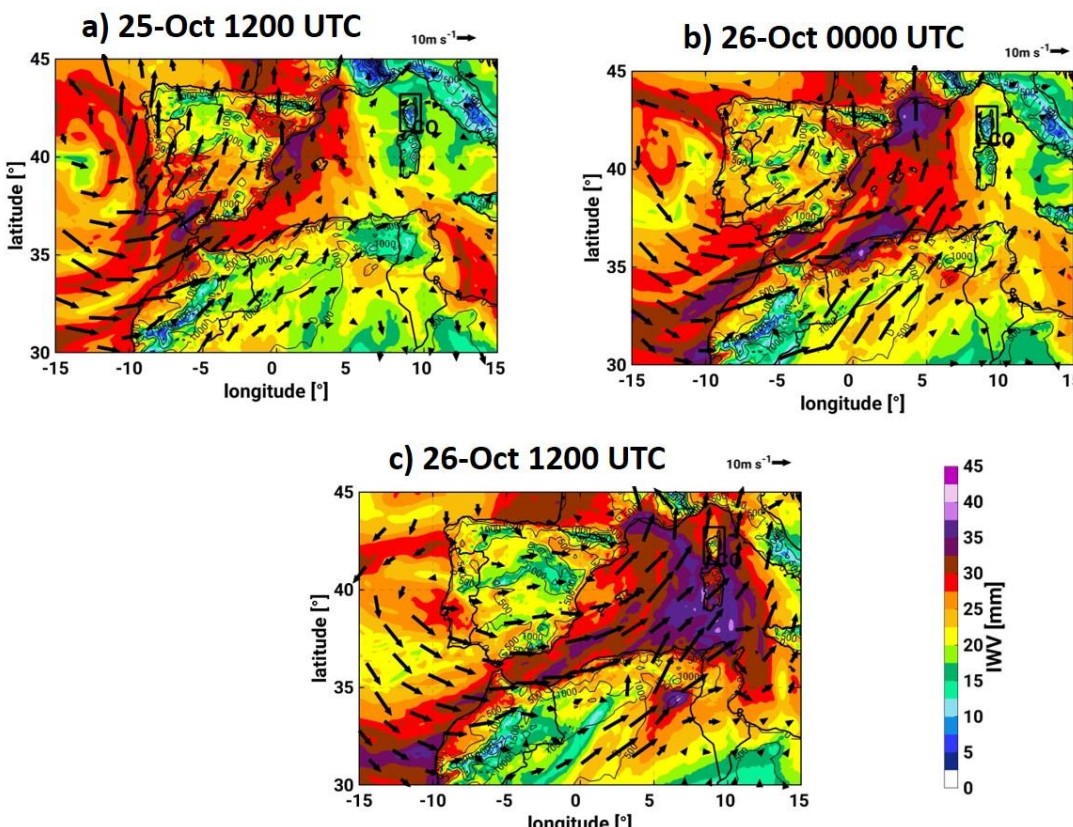

**Figure 4. Spatial Distribution of Integrated Water Vapour (IWV) and winds at 850 hPa as represented by CTRL-7 on the 25-Oct-2012 at 1200UTC (a), 26-Oct 0000UTC (b) and on the 26-Oct-2012 at 1200UTC (c).**




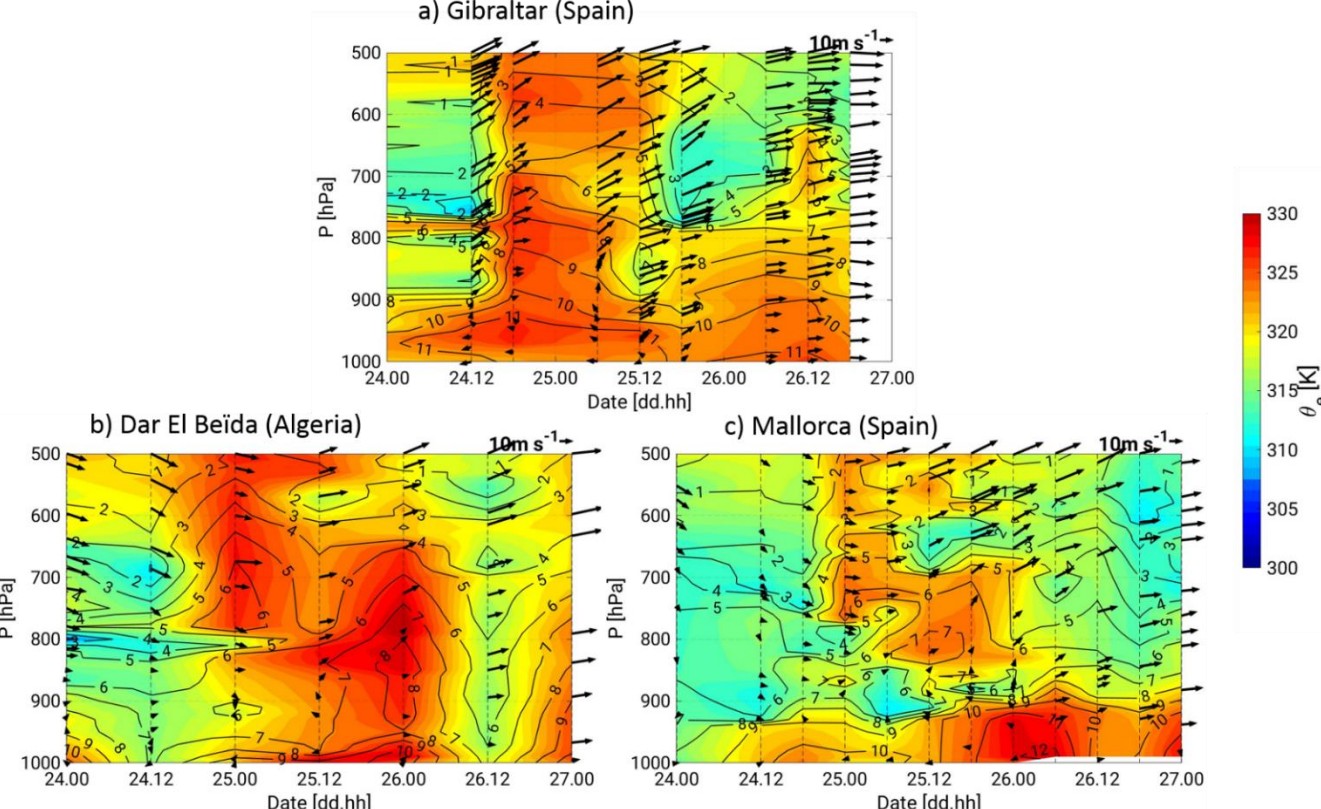


**Figure 5. Height-time cross-sections at Gibraltar (a), Dar el Beïda (b) and Mallorca (c) between 24-Oct 0000UTC and 27-Oct 0000UTC. The colour shading stands for Equivalent Potential Temperature ($\theta_e$), the contours denote specific humidity and the arrows show the direction and speed of horizontal wind at the stations. The location of the radiosondes can be seen in Fig. 1.**

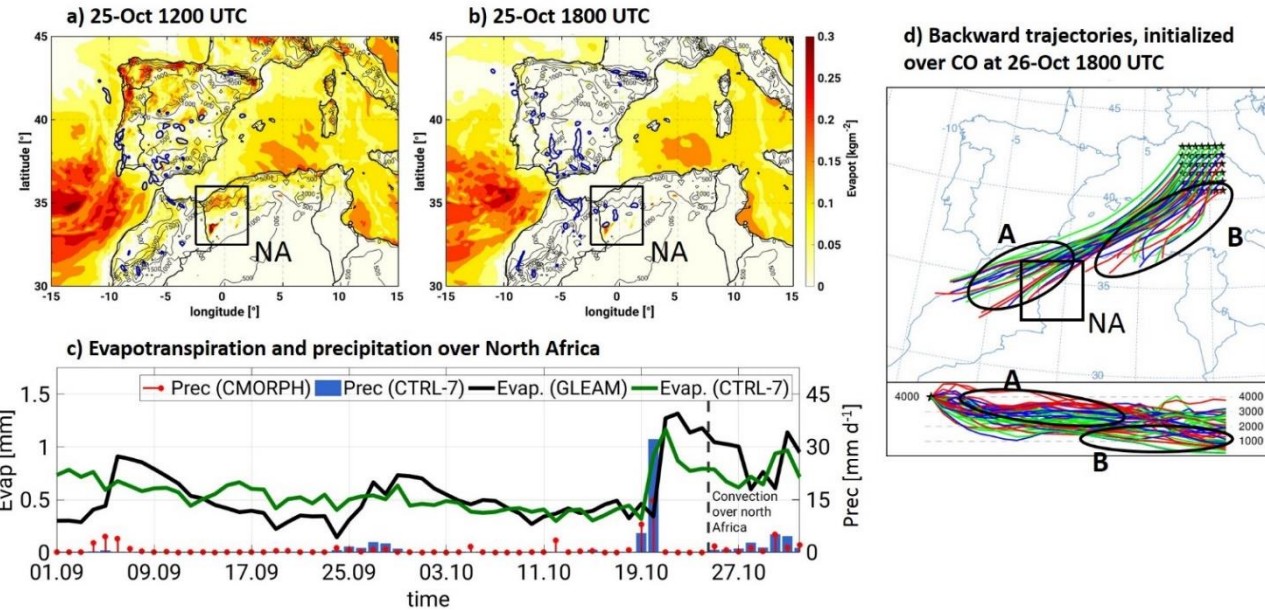


**Figure 6.** Spatial distribution of evapotranspiration (colour shading) and vertical updrafts with wind speeds larger than 0.25 m/s (blue contours) simulated with CTRL-7 on the 25-Oct at (a) 1200UTC and at (b) 1800UTC. The black box denotes the region of intense evaporation between 20-Oct and 27-Oct. (c) Spatially averaged evapotranspiration (GLEAM and CTRL-7) and daily precipitation (CMORPH and CTRL-7) over north Africa (black box). The CTRL-7 precipitation has been upscaled to the coarser grid of CMORPH (0.0727°). Likewise, CTRL-7 evapotranspiration has been upscaled to the coarser grid of GLEAM of 0.25 °. (d) Lagrangian backward trajectories obtained with the HYSPLIT model, starting on the 26-Oct-2012 1800 UTC (initiation of the event over Corsica) back to for 24 h



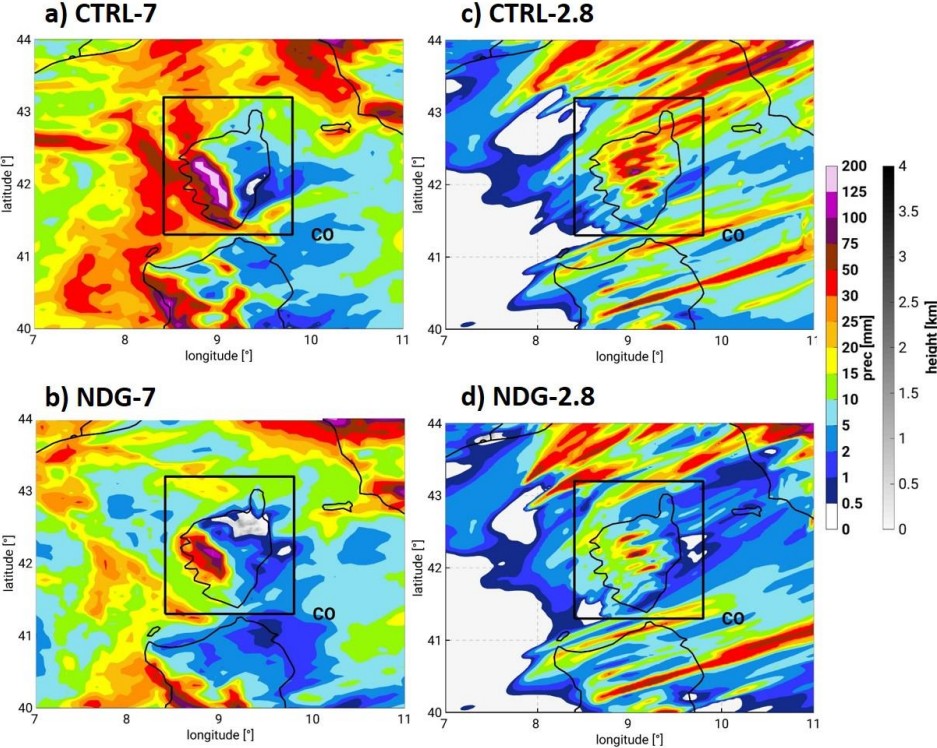

**Figure 7. COSMO-CLM accumulated precipitation over Corsica between 26-Oct 1300 UTC and 27-Oct 1500 UTC i.e. during the period of precipitation over the island.**

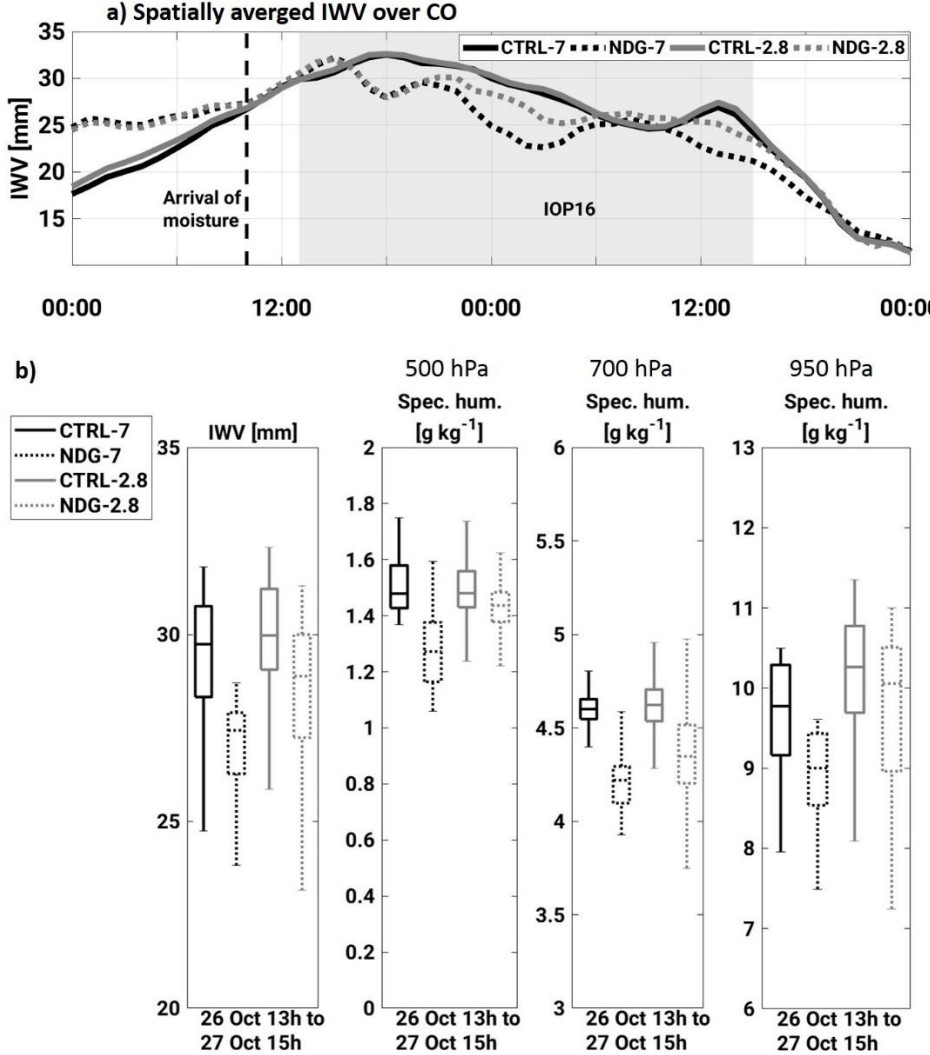

**Figure 8. Spatially averaged IWV for all simulations during the event. The area for averaging is shown in Fig. 1 (CO) and the model output has been upscaled to a common coarser grid. The period shown is 26-Oct 0000 UTC to 28-Oct 0000 UTC. (b) Box and whiskers plots showing the median, the percentiles 25 and 75 and the extreme values of IWV and specific humidity at 500 hPa, 700 hPa and 950 hPa. All box and whiskers are obtained from the distribution of values for the shown quantities between the 26-Oct 1300 UTC and 27-Oct 1500 UTC over the study region CO.**


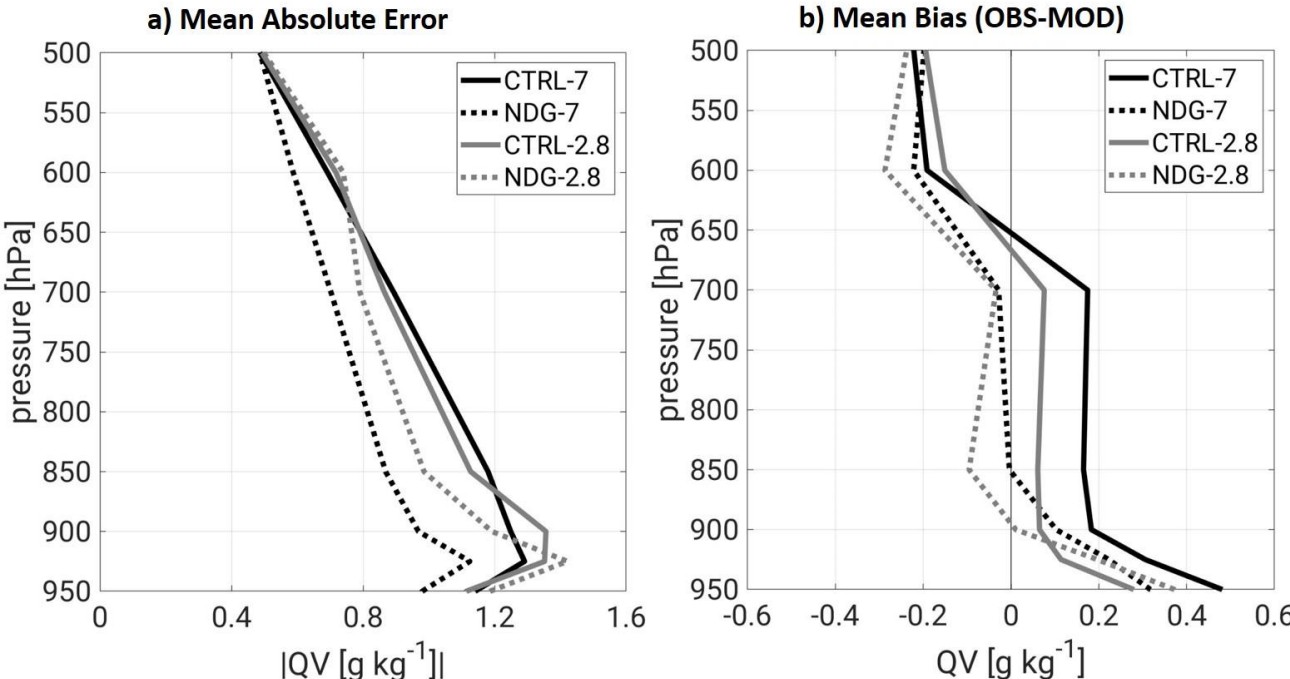


**Figure 9. Validation of the model representation of the vertical distribution of specific humidity quantified by the Mean Absolute Error (a) and the Mean Bias as the OBS-MOD (b). All model values are validated against the radiosondes available in the period 26-Oct 0000 UTC to 28-OCT 0000 UTC at the 7 stations within the 2.8 km simulation domain (see red box in Fig. 1). All model values have been interpolated to the location of the radiosonde station from the nearest neighbours and the vertical specific humidity values have been interpolated to eleven pressure levels between 950 and 500 hPa. No values are shown below 950 hPa given the lack of available data for several stations below this level.**



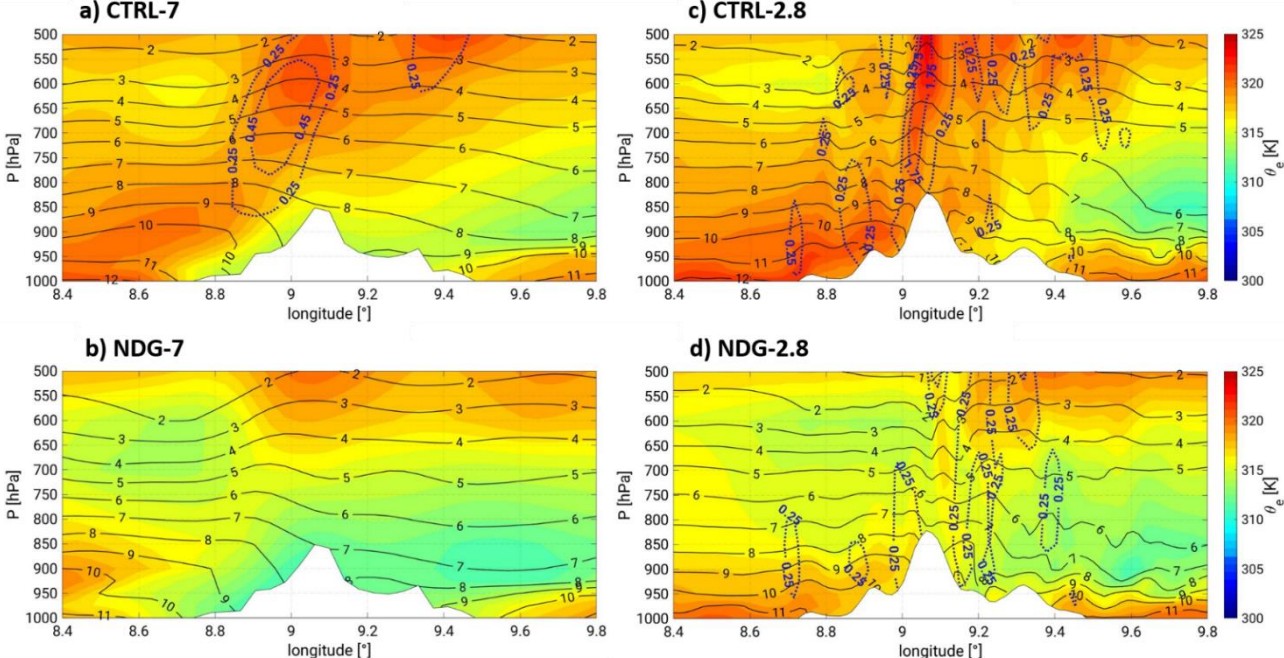

**Figure 10. Vertical cross sections along the mean wind direction between 700 hPa and 1000 hPa (see transect in Fig. 1) on the 26-Oct 1700UTC. The selected hour corresponds to the first precipitation maximum represented over Corsica by the COSMO simulations. The position of the transect is represented in Fig. 1. The specific humidity values are represented as black contour lines, the colour shading represents equivalent potential temperature ($\theta_e$). Vertical windspeed is represented in blue dashed contours, where isolines start at 0.25 m/s.**

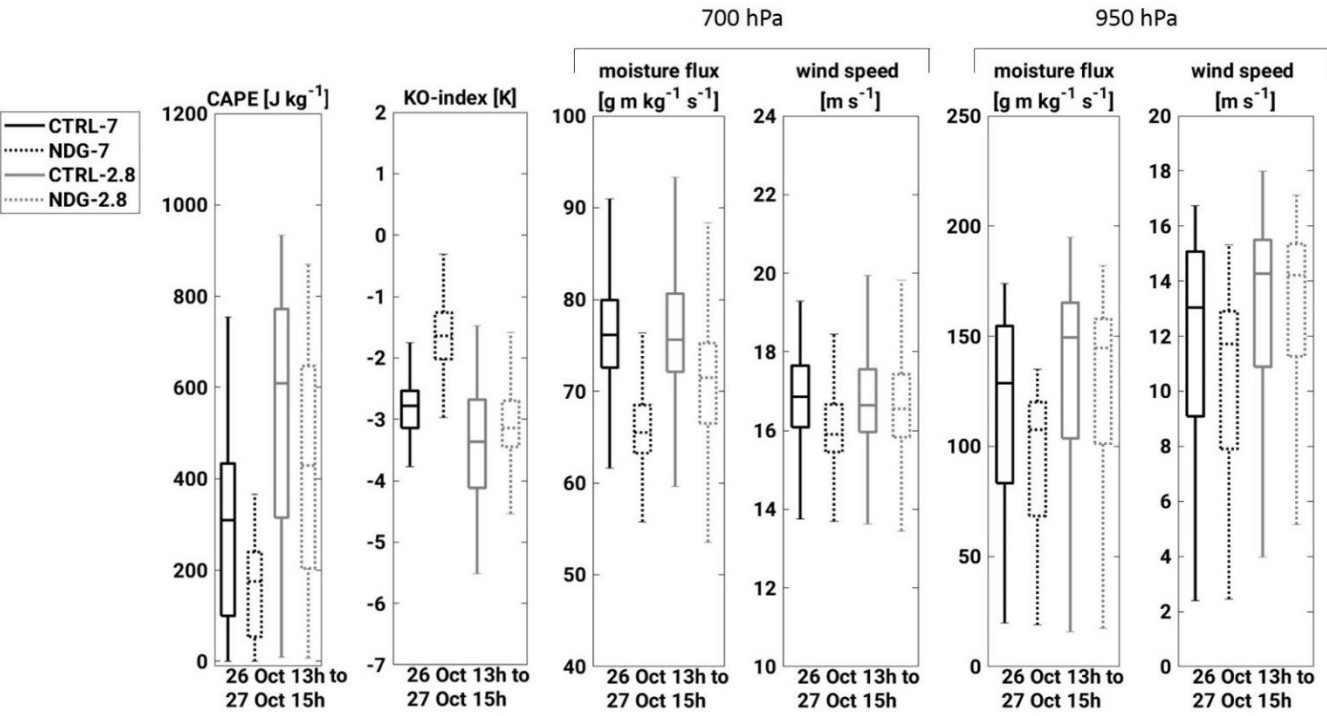

**Figure 11. Box and whiskers plots showing the median, the percentiles 25 and 75 and the extreme values of CAPE-ML, KO-index, moisture flux and wind speeds. All box and whiskers are obtained from the distribution of values for the shown quantities between the 26-Oct 1300 UTC and 27-Oct 1500 UTC over the study region CO.**


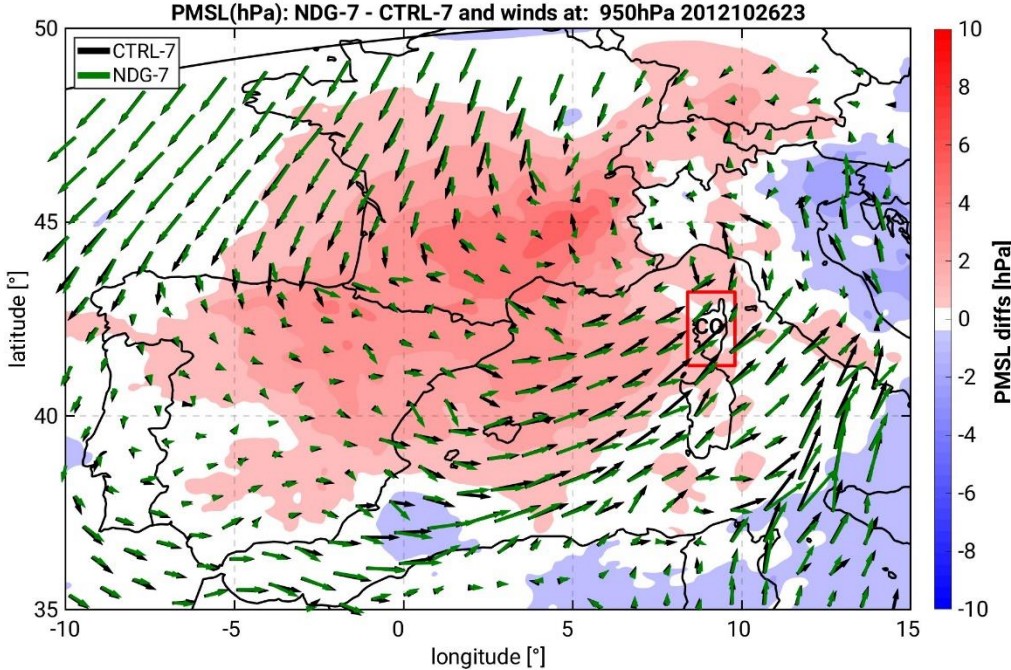

**Figure 12. Spatial distribution of the differences in Pressure at the Mean Sea Level (PMSL) between NDG-7 and CTRL-7 on the 26-Oct 2300 UTC. Horizontal winds at 950 hPa are represented by black (CTRL-7) and green (NDG-7) arrows.**
