# Peer review of "Assessing Atmospheric Moisture Effects on Heavy Precipitation During the HyMeX IOP16 Using GPS Nudging and Dynamical Downscaling"

_Natural Hazards and Earth System Sciences, 2019_

## Referee Comment (RC1) · Anonymous Referee #1 · 2 Jan 2020

**1    General comments**

The manuscript presents a study of heavy precipitation in the Mediterranean area. Different data sources are used to characterize the event in terms of moisture. Among them, COSMO-CLM runs play a central role.  GPS-ZTD data are assimilated in COSMO-CLM through nudging in nested domains of 7 km and 2.8 km, respectively. The authors conclude that nudging GPS ZTD data improves the modelling of precipitation for this event.

As correctly pointed out by the authors, the role of moisture in heavy precipitation events is of great interest. Unfortunately, the discussion paper is burdened with substantial flaws.

First of all, it contains numerous purely speculative assertions that are not at all substantiated. This is reinforced by the fact that results are often given before they are demonstrated (this should be reversed). More rigorous justifications are needed to make the discussion paper a compelling study.

Second, as justly stated by the authors at the end of their concluding section, their study relies on one case only. This undermines the results concerning the sensitivity study to the assimilation of GPS ZTD data, all the more since the different runs start much earlier than the event itself. In doing so, it is difficult to link the impact of the assimilation of ZTD-GPS data to the simulated precipitation. The differences observed during the chosen event might simply be due to the chaotic nature of the atmosphere. In any case, such a configuration does not help understand where and when the assimilation of GPS ZTD data has the most impact on precipitation. For example, the authors should include some estimation of the impact duration of the GPS-ZTD data assimilation by running some free runs and determining when they converge, or start the different runs shortly before the event starts.

Third, I deem it doubtful that the "modelling experiments demonstrated the benefit of sub-hourly GPS-ZTD nudging to improve the modelling of precipitation" for the following two reasons. First, the benefit of a sub-hourly frequency versus, e.g., a hourly frequency, has not been demonstrated. Second, there is no "pure" modelling in the study since the model is constantly perturbed by the modification of its moisture field. In doing so, it is highly speculative to interpret any physical process in a consistent way. The authors should be more specific about the usefulness of continuously nudging moisture.

More details regarding these general comments are given below.

**2 Specific comments**

**L47-49:** The authors point out the interest of assimilating humidity data at sub-hourly frequencies. I suggest the authors study the sensitivity of the assimilation frequency by carrying out an additional experiment with a one-hour assimilation frequency. This would demonstrate to what extent a sub-hourly assimilation frequency is needed.

**Case study and numerical set-up:** Why run experiments that last several months and study one case only? The differences seen in this specific case could be caused by a lower predictability rather than to improvements in the description of the humidity field. Why is there no NDG-2.8 simulation forced by NDG-7? In theory, shouldn't this configuration yield the best results?

**L246-247:** Can the authors please elaborate on why, "under a weak synoptic forcing, the impact of the GPS-ZTD is larger given the strongest correction of the lower to middle tropospheric humidity"?

**Section 3.3:** This section contains general statements, which are true, but are not new: writing that moisture is swept by a front and/or originates from the seas and oceans is true, but it would be much more interesting to know which fraction is swept and which fraction comes from evaporation. Moreover, many statements in this section are not properly backed. The CTRL-7 simulation is validated later (that would be better here or before!), no moisture budgets are computed, some HYSPLIT backward trajectories are computed, but the underlying analyses (GDAS) are not validated. A validation against MODIS is mentioned but not

shown(?!) In addition to the lack of GDAS validation for this event, I wonder what 0.5 degree resolution parcels represent. Furthermore, I suspect that labels A and B are inverted in the lower panel of Figure 6.b. If I am wrong, it means that I don't understand this panel.

**L326-328:** The specific humidity in Figure 5.a is said to be in agreement with the simulated IWV in Figure 4.a. Why not compute the IWV from the radiosonde to quantify it?

**L352-353:** Why does this "further promote the moisture uptake form the Sea"?

**L396-398:** Looking only at the maximum precipitation value is not really fair because the rain gauge network is rather sparse compared to the scale of the studied phenomenon (and simulated maximum value). At first glance, I would say that CTRL-2.8 is better than NDG-2.8. A proper validation of numerical simulations against rain gauges is needed.

**L400:** The authors do not really assess the accuracy of model moisture outputs: since most of the radiosonde locations are near GPS receivers, they rather assess the accuracy of GPS IWV retrievals. This most certainly explains why NDG-7 and NDG-2.8 results are so close to each other. To really assess the accuracy of model moisture outputs, free runs should be evaluated.

**L411:** Figure 8.a shows the IWV over the CO domain, not Corsica. This is important, because the most noticeable differences among the nudging simulations may be over the sea since IWV mainly comes from GPS receivers on ground. The authors should adapt the rest of their interpretation of Figure 8.a accordingly.

**L423:** Where is it evidenced that the humidity reduction takes place below 500 hPa?

**L429-447:** A validation against radiosondes is presented. Which radiosonde profile has an impact on the event under consideration? They are all located either east,

north, or south of Corsica, while the authors showed that moisture comes from the (south-)west.

**L466-467:** Where is a "decrease of humidity close to ground" shown?

**L469-471:** A simulation nudging IWV at 2.8 km and forced by NDG-7 would be useful here.

**L487:** Why is the effect of the low-pressure system change exclusively seen in the 7 km simulations? Isn't the wind changed in the 2.8 km simulations, too?

**L568-569:** The high-temporal resolution of GPS-ZTD observations has not been shown to facilitate a better representation of the water vapour variability and a better regulation of the accumulated precipitation. To do this, different temporal resolutions should have been used.

**3  Technical corrections**

**Title:** "Effects ... using" something does not really mean anything. I suggest adding "Assessing", "Evaluating" or some other relevant term to amend it.

**L68-69:** What is the "location" of a convective system? In convective updrafts, saturation is likely to be reached. To make it meaningful, the sentence "Khodayar ... (HPE)." needs more context.

**L130:** It is already written (L100 and 103) that the GPS data come from 25 European and African networks with over 900 stations.

**L217:** Is it useful to describe the vertical interpolation in the nudging scheme when it comes to assimilate GPS data?

**L230-231:** It is unclear. What is the "iterative process"? If Eq 2 is used, it is not an iterative process, is it?

**L241:** It is stated that the 7 km runs are forced by ECMWF analyses. How often?

**L244:** The terms "large precipitation reductions" is unclear. The authors must specify which experiences they are referring to.

**L254:** First occurrence of "$\theta e$": it should be expanded here, not in L273.

**L298:** What is an "offshore-size" convective system?

**L367:** "Ellipse A in Fig. 7" or in Fig. 6.d?

**L369:** "Ellipse B in Fig. 7" or in Fig. 6.d?

**L447:** What is an "accuracy rate"?

**L454:** What does "height-surface" mean? Do the authors simply mean "vertical"?

**L455:** What does "vertical-horizontal direction" mean?

**L476:** "Wind speed" seems to refer to the horizontal wind speed. This should be specified right away.

**Figure 3:** SEVIRI is not a satellite, it is an instrument.

**Figure 6:** Units of evapotranspiration are either kgm$^{-2}$ or mm. Over what period of time? In Figure 6.d, the two sets of trajectories cannot be distinguished. Is it possible to plot each of them with a distinct color? I suppose the lower panel is a time series. Is it possible to add labels in the x-axis?

---

## Referee Comment (RC2) · Anonymous Referee #2 · 9 Feb 2020

**Review of the paper:**

Atmospheric Moisture Effects on Heavy Precipitation During the HyMeX IOP16 Using GPS Nudging and Dynamical Downscaling

By: Alberto Caldas-Alvarez, Samiro Khodayar

**General comment**

This paper presents the impact of assimilating GPS-ZTD in the COSMO-CLIM model focusing on the case study of HyMeX-IOP16. The first part of the paper shows the physical processes responsible for the heavy precipitation of the case study, while the second part focuses on the impact of assimilating realistic humidity observations  on the hindcast of the event. The impact of evapotranspiration over North-Africa for this event is shown, even if not quantified.

The paper is well written and presents an interesting subject. Nevertheless, there are points of the paper that can be improved and sometime clarification is needed. The major and minor points are reported below.

**Major points**

1) The authors must clarify that this is a diagnostic study and not prognostic. The impact of assimilating GPS-ZTD is quantified by comparing two simulations: the first doesn't assimilate GPS-ZTD, while the second assimilates GPS-ZTD continuously. While this is an important comparison, it must be clarified that the paper doesn't assess the role of GPS-ZTD in a prognostic approach for the case study. Also the importance of sub-hourly data assimilation is not shown. To do that a comparison between two simulations one assimilating GPS-ZTD on a hourly basis and the other one assimilating GPS-ZTD every 10 minutes (as in the paper) should be performed. However, I understand that this requires adding new simulations, which can be avoided deleting the sentences where the importance of sub-hourly assimilation is emphasised.

2) In the section 3.3 emphasis is given to the transport of humidity from North Africa for the event. It would be interesting to give a comparison between this source of moisture and that coming from the western Mediterranean Sea to define better this contribution.

3) Considering the nudging scheme there is no information on the parameters of the Second order autoregressive function. How they are determined? Line 217 has a comment on the vertical adjustment that doesn't apply to the specific case. It also unclear how the $q_v$ profile is constructed iteratively (Lines 230-232). Do you mean that it is modified by nudging until a difference is attained or something different?

**Minor points**

The e-mail for correspondence seems wrong.

Line 28: there is a "." after "Additionally", while a comma is expected.

Line 33: During heavy precipitation events, rain rates can be much higher than 20 mm/h.

Line 47: Check the ";".

Line 188-189: Please revise the English, specifically "Where".

Line 244: Check the sentence "given the large precipitation reduction". Do you mean when you assimilate GPS-ZTD?

Line 248: I suggest giving more details about the Agreement Index (AI).

Line 271: "the low-level flow …"

Line 426: the number for the 2.8 km are wrong. Check.

Line 523: Check the sentence (there is also a typo error).

Line 577-578: This sentence is rather unclear. It is important to note that, in general, the adjustment introduced by GPS-ZTD could be a function of the height if variational approaches are considered, through the background error matrix.

---

## Author Comment (AC1) · 22 Mar 2020

Answers to reviewer 1 (Paper-IOP16-NHESS)

We would like to thank the anonymous reviewer for the valuable comments, suggestions and questions. We have considered all comments. We believe that the quality of the manuscript has increased, thanks to the comments of the reviewer and acknowledge how answering the raised questions has been crucial for a better exposition of our key messages.

[Figure]

**1 - GENERAL COMMENTS**

R1C1. First of all, it contains numerous purely speculative assertions that are not at all substantiated. This is reinforced by the fact that results are often given before they are demonstrated (this should be reversed). More rigorous justifications are needed to make the discussion paper a compelling study.

We have corrected all speculative answers and hypothesis in the manuscript, as well as reorganized the conclusions to be given after presenting the corresponding results only.

We would like to add, that this case study belongs to a series of modelling experiments performed in our group which are object of an on-going publication. In these modelling experiments we analysed the impact of nudging GPS-ZTD on several heavy precipitation events of the HyMeX period as well as the statistics of the complete autumn season. This is why at times in the manuscript some statements are introduced which are not justified. These statements arise from further analysis on the period which are not presented in the paper. This has been corrected accordingly to show only information relative to IOP16.

R1C2. Second, as justly stated by the authors at the end of their concluding section, their study relies on one case only. This undermines the results concerning the sensitivity study to the assimilation of GPS ZTD data

We have adapted the manuscript to clearly state that the withdrawn conclusions correspond to the presented case only (IOP16). This is done for example at lines L579-580. In connection with the previous answer, we will submit in the coming weeks a publication with analyses on heavy precipitation and the impact of the GPS-ZTD nudging for the whole season. The selected case study object of this paper is a relevant event of the period since the GPS-ZTD showed very interesting impacts on convective development, as explained in the manuscript. The simulations presented in this paper are used, together with other simulations and observational data, in the analyses of this

future publication on the seasonal time scale.

To let the readers know about the broader scope of these studies we have adapted the following section.

2.2.2 The GPS-ZTD Nudging Sensitivity Experiments (L244-246)

IOP16, the case study of this paper, is one of them which is especially interesting given the large reduction of maximum precipitation (-20 %) induced by the GPS-ZTD nudging over the investigation area of Corsica in the course of 26 h. The remaining cases of the autumn 2012 period and analyses on the complete season are part of a series of modelling studies from our working group, including a PhD thesis by Caldas-Alvarez (2019). IOP16 is also suitable to assess the benefit of atmospheric moisture corrections with GPS-ZTD nudging given the important role of the local orographic and instability factors in triggering and maintaining convection rather than the large-scale upper level forcing.

R1C3. all the more since the different runs start much earlier than the event itself. In doing so, it is difficult to link the impact of the assimilation of ZTD-GPS data to the simulated precipitation. Differences observed during the chosen event might simply be due to the chaotic nature of the atmosphere. In any case, such a configuration does not help understand where and when the assimilation of GPS ZTD data has the most impact on precipitation. authors should start the different runs shortly before the event starts.

This is in connection with the previous answers. We believe that analysing one case study from a seasonal simulation can be useful to relate the findings for this one case study with the analysis of the whole Autumn period. In addition, we think that starting the simulation 50 days before the event allows us to study the hypothesis that correcting the atmospheric moisture also during previous events (not only during the selected case study) would bring a larger improvement for precipitation. In this sense we wanted to profit from the availability of the unique GPS-ZTD data set, that covers the whole

[Figure]

SOP1 period and part of 2013 (1-Sep-2012 to 20-Nov-2013). Moreover, the earlier initialization introduces a spin up time for slower processes such as soil-atmosphere interactions that are affected by precipitation during previous events.

Since we consider relevant the question raised by the reviewer, how different the impact will be depending on the initialization day, we performed additional simulations starting only a few days before the event using the same settings and forcing data but initialized on the 20-Oct-2012. This date was selected to ensure the representation of the large moisture evapotranspiration over north Africa (20 to 21-Oct), demonstrated in Fig. 6 of the manuscript.  

Figure 1. Spatially averaged IWV (a, c) and Precipitation (b, d) for the 7 km simulations (a, b) and the 2.8 km (c, d) during the event. The area of the spatial averages is Corsica. The model output has been upscaled to a common coarser grid of 8 km to allow for comparison. The period shown is 26-Oct 0000 UTC to 28-Oct 0000 UTC.

Regarding IWV, the shorter simulations show differences in that the IWV increase happens 1 to 2 h earlier in the CTRL runs initialized on the 20-Oct (Init20Oct). This holds for both resolutions. The NDG runs show, however, no perceptible differences for the temporal evolution of spatially averaged hourly IWV. On the contrary, precipitation shows relevant differences for both resolutions (Fig. 1b and Fig.1.d). NDG-7-Init20ct shows two maxima instead of three with somewhat larger precipitation intensities. NDG-2.8-Init20Oct shows a delayed (1 h) onset of precipitation and lower precipitation rates as compared to CTRL-2.8-Init1Sep.

A plausible explanation for these findings, as stated by the reviewer, is the chaotic nature of the atmosphere that is affected by all perturbations in earlier stages. This analysis will be added in the annex of the manuscript as supplementary information for the readers. We prefer not to include it in the body of the paper, since it deviates from the objective for the article.

We believe using the longer simulations, i.e. started on the 1-Sep-2012, have the

advantage of relating the findings with those of our planed publication on the complete autumn season and that they benefit from the whole duration of the GPS-ZTD data set. Therefore, we will present in our manuscript the analyse with these simulations as in the first version but will add a supplementary subsection within 4-Nudging Effects on Convection where the sorter simulations will be introduced and analysed. This will be done to present the results of the 1h vs. 10min frequency comparison (see comment R1C5).

R1C4. authors should include some estimation of the impact duration of the GPS-ZTD data assimilation by running some free runs and determining when they converge

Our simulations span the whole autumn season to cover the duration of the HyMeX SOP1 and the availability of the GPS-ZTD data. This is why it is not possible to answer when would the assimilated and the free runs converge. Our scientific question was how is convective precipitation impacted by a continuous correction of IWV in the model? The modelling set up was planned in a different way to usual data-denial assimilation experiments (Borderies, et al., 2019; Benjamin, et al., 2010; Mahfouf, et al., 2015) in that these experiments data assimilation is performed during a given time window to obtain the initial conditions for a future step of first guesses. These data-denial experiments allow to study how long the impact of an observation type stays in the system as the analysed runs converge with the forecasts, but our set-up was not conceived to provide information on this.

R1C5. the benefit of a sub-hourly frequency versus, e.g., at hourly frequency, has not been demonstrated (coming from my sentence "modelling experiments demonstrated the benefit of sub-hourly GPS-ZTD nudging to improve the modelling of precipitation") Thanks for this comment. We acknowledge that the sentence was misleading. We wanted to express that our experiments nudging GPS-ZTD with a 10-min frequency showed an improvement for this case study. This has been corrected in the manuscript.

We agree that this is an interesting aspect, thus we have performed supplementary

simulations to investigate more in detail this issue. We run the 20-Oct-2012 to 28-Oct-2012 0000 UTC period as in the comment R1C3, applying the GPS-ZTD nudging with a frequency of 1 h, as opposite to the 10 min frequency. All other settings are the same as NDG-7-Init20Oct and NDG-2.8-Init20Oct respectively. Hereafter we name these simulations simply as CTRL-7, CTRL-2.8, NDG-7, NDG-2.8, NDG-7-1h and NDG-2.8-1h. These six simulations have been initialized on the 20-Oct-2012  

Figure 2. Spatially averaged IWV (a, c) and Precipitation (b, d) for the 7 km simulations (a, b) and the 2.8 km simulations (c, d) during the event. The area for averaging is Corsica. The model output has been upscaled to a common coarser grid of 8 km. The period shown is 26-Oct 0000 UTC to 28-Oct 0000 UTC. The comparison is between the runs using a temporal nudging frequency of 10 min (NDG-7, NDG-2.8) against nudging at a temporal frequency of 1h (NDG-7-1h, NDG-2.8-1h).

The results show no differences on the temporal evolution of IWV. This holds for 7 km and 2.8 km. This can be explained by the fact that we calculate the spatially averaged IWV at sharp hours (i.e. 0000 UTC, 0100 UUTC, 0200 UTC, etc.), precisely is at those times when the GPS-ZTD data is assimilated in the NDG-7-1h and the NDG-2.8-1h runs.

For precipitation, there is a slight impact for the 7 km runs, but not for the 2.8 km. The NDG-7-1h simulation shows a somewhat larger precipitation than NDG-7 at 2000 UTC on the 26-Oct (Fig. 2b) corresponding to an increase from 30 mm to 50 mm at the western shore of the island (Fig.3.b and Fig.

Figure 3. COSMO-CLM accumulated precipitation over Corsica between 26-Oct 1300 UTC and 27-Oct 1500 UTC i.e. during the period of precipitation over the island and RG.

To delve further into which aspects of precipitation representation have been improved, we present in Table 1 further validation metrics using the Rain Gauges (RG) as reference.

Table 1 shows the RMSE of the anomalies of hourly precipitation rates (first column), the differences (OBS-MOD) of the standard deviations of hourly precipitation (second column) and the spatially averaged differences of accumulated precipitation during the whole event, i.e. between 26-Oct 1300 UTC and 27-Oct 1500 UTC (third column). The last three metrics are obtained after interpolating the COSMO-CLM precipitation values to the location of the RG stations. On the other hand, columns four and five of Table 1, show differences of the standard deviation and maximum value of precipitation for COSMO-CLM over land without interpolation. That means, we have obtained all 27h-accumualeted precipitation values simulated by COSMO-CLM over land and have calculated the standard deviation and the maximum. We have done the same for all RG measurements and the differences are shown. We do this to avoid double-penalty problems due to a possible misrepresentation of the maxima location (Wernli, et al., 2008; Gilleland, et al., 2009). The formulas used are included in Table 2.

Table 1. Metrics of the precipitation validation against RG. The model precipitation has been interpolated to the location of the RG for the first three columns and all precipitation values simulated by COSMO-CLM over the island of Corsica are used in the last two columns. This is done to avoid double-penalty problems due to a shifting of the precipitation maxima. N is the number of RG stations and M the total number of grid points over land. The units are mm.

Table 2. Precipitation validation metrics.

Overall, we see that nudging GPS-ZTD data is beneficial for the 7 km grid with little difference between nudging with 1h frequency or 10 min. If any, we see a slight advantage in nudging GPS-ZTD data with 10 min for the representation of the hourly standard deviation rates. The same holds for the 2.8 km, assimilating with a 10min frequency shows very weak differences with respect to assimilating with 1h frequency.

These results will be included as an additional subsection in section 4.1 (Nudging effects on Precipitation).

R1C5. there is no "pure" modelling in the study since the model is constantly perturbed by the modification of its moisture field. The authors should be more specific about the usefulness of continuously nudging moisture.

We decided to perform a continuous nudging to study the scientific question "how does simulated precipitating convection respond to a sub-hourly moisture correction?". In the following version of the manuscript we state more clearly that our study is diagnostic and that we do not provide an assessment of the prognostic use of GPS-ZTD nudging. We have corrected the following parts.

In the Abstract "In this study, we use a diagnostic approach to assess the sensitivity of precipitating convection and underlying mechanisms during a heavy precipitation event (HyMeX intensive observation period 16) to corrections of the atmospheric moisture spatio-temporal distribution."

In Section 2.2.2 of the Methods. The GPS-ZTD Nudging Sensitivity Experiments "The Nudging scheme is used to assimilate GPS-ZTD data to assess the sensitivity of heavy precipitating convection to corrections of the spatio-temporal distribution of atmospheric moisture. We use a diagnostic approach as opposite to commonly use data-denial experiments." As stated earlier in this document, our modelling setup differs from other data-denial assimilation experiments for instance (Benjamin, et al., 2010; Borderies, et al., 2019; Sahlaoui, et al., 2019). Hence the idea of finding new initial conditions for subsequent forecast intervals through data assimilation does not apply in our experiments.

We believe, our study is valuable we were able to show the potential of such observation nudging since, for this case study, hourly RMSE values, the maximum and standard deviation were improved using the GPS-ZTD nudging and in the 7 km grid. Furthermore, we also observed the problems of COSMO in representing the moisture vertical gradient with substantial differences between 7 km and 2.8 km and how the GPS nudging could not correct sufficiently the vertical humidity errors. We saw that

for this case study, the reduction of instability and of humidity at the free-troposphere exerted the largest control for convective precipitation. The fact that we nudge our simulations to GPS values continuously for a whole season has the advantage that the corrections introduced are not only present during the event (as usually in data-denial experiments) but also in past events.

2 - SPECIFIC COMMENTS

R1C6. L47-49: The authors point out the interest of assimilating humidity data at sub-hourly frequencies. I suggest the authors study the sensitivity of the assimilation frequency by carrying out an additional experiment with a one-hour assimilation frequency. This would demonstrate to what extent a sub-hourly assimilation frequency is needed.

See section 1.

R1C7. Case study and numerical set-up: Why run experiments that last several months and study one case only? The differences seen in this specific case could be caused by a lower predictability rather than to improvements in the description of the humidity field.

See section 1.

R1C8. Case study and numerical set-up: Why is there no NDG-2.8 simulation forced by NDG-7? In theory, shouldn't this configuration yield the best results?

We decided to force the NDG-2.8 simulations with the CTRL-7 simulations to be able to compare NDG-2.8 to CTRL-2.8 directly. By doing so we were able to assess the direct impact of the GPS-ZTD nudging at the 2.8 km resolution which would have not been possible if the NDG-2.8 runs had been forced by different boundary conditions than their CTRL counterparts.

R1C9. L246-247: Can the authors please elaborate on why, "under a weak synoptic forcing, the impact of the GPS-ZTD is larger given the strongest correction of the lower

to middle tropospheric humidity"? Thanks for this comment. We acknowledge that the information of this sentence is not fully explanatory. This conclusion arises from the results we obtained in the seasonal simulations of the complete autumn period. Given the large number of heavy precipitation cases during that season and the advantage of having nudged GPS data continuously, we were able to ascertain which cases were impacted the most by the GPS-ZTD nudging. Precisely we found a larger sensitivity in the model for those cases of weak synoptic forcing such as IOP16 where the role of the local factors (i.e. latent instability, low-level moisture, wind convergence and orographic triggering) is more important for convection than the direct forcing of the large scale.

To clarify this question, we include here results of the analysis on all cases of the Autumn 2012 period. These graphs will not be included in the new version of the manuscript as they would be outside the scope of the paper but will be included in a future publication from our working group. We include them here for clarification of the raised question. Finally, in the new version of the abstract this point will be adequately explained and referenced.

Figure 4. Scatterplot of IWV and precipitation variations due to the GPS-ZTD nudging for all events in the 1-Sep to 20-Nov period for the 7 km (a) and the 2.8 km (b). Each, dot accounts for one event. The events are detected by averaging hourly precipitation values over 8 study regions, of the HyMeX campaign (Ducrocq, 2015) and selecting those with average precipitation reaching 0.4 mm/h. The differences in precipitation and IWV ($\Delta$Precip, $\Delta$IWV) are obtained after subtracting the average precipitation of NDG to CTRL. The degree of synoptic forcing (strong, weak, very weak) is calculated following the convective-adjustment time-scale criteria of Keil et al. (2013). The gray-scale color bar shows the duration of the event in the CTRL-7 and CTRL-2.8 runs respectively. The yellow shaded areas at the sides denote the areas of the upper and lower quartiles of precipitation variations. The upper-left and upper-right text boxes show statistics on the shown events.

In Fig. 4 we show the differences of IWV and precipitation for the events of the Autumn

2012 period between CTRL and their NDG counterparts (NDG-CTRL). We can see that 67 and 84 events took place in the total of the 8 investigation areas of the HyMeX campaign in the 7 km and the 2.8 km runs respectively. We can see, that 63 % of the cases in the 7 km runs and 75 % in the 2.8 km where categorized as synoptically strongly forced by the convective adjustment time scale criteria, being the rest of them either weakly or very weakly forced. The events lying in the upper and lower quartiles of the precipitation distribution (yellow shaded areas) show that the most impacted events where those of a weak synoptic forcing (55 % in the 7 km runs and 61 % in the 2.8 km runs). The 2.8 km runs also show two out of three cases of very weak synoptic forcing within the upper and lower quartiles.

R1C10. Section 3.3: This section contains general statements, which are true, but are not new: writing that moisture is swept by a front and/or originates from the seas and oceans is true, but it would be much more interesting to know which fraction is swept and which fraction comes from evaporation We agree with the reviewer that a quantification of the different terms of Evaporation and moisture flux over the investigation area NA and the Mediterranean would be most interesting. We have obtained the terms, described in Lamb et al. (2012) over the investigation areas NA (North Africa) and MED (Mediterranean Sea) for this purpose, see Fig. 5.a.

The calculation of these terms entails simplifications for example, of the turbulent and microphysical processes that introduce relevant uncertainties. Hence, what we provide here is an estimation of the different contributions.

$$\Delta IWV = E + (-P) + MFC \quad (1)$$

All terms of Eq. 1 are expressed in mmh-1, where positive signs of the Evaporation (E) and Integrated Moisture Flux Convergence (MFC) imply an increase of Integrated Water Vapour variations ($\Delta IWV > 0$) within the NA and MED volumes. On the contrary, if precipitation and water vapor divergence occur (MFC<0) IWV decreases ($\Delta IWV < 0$). The volumes cover the areas in Fig. 5.a where the integrations of IWV and MFC are

performed from the first to the last model levels.

Fig. 5.b shows similar information to Fig.6.c of the manuscript. Intense precipitation occurs over NA on the 20-Oct-2012 with the subsequent decrease of IWV, and intense evaporation over the area on the 21-Oct and 22-Oct at midday. This is the moment when solar radiation is strongest and evaporation is intensified over this wet soil. Please note the change in the axis scales between the different panels of this figure and those of Fig.6.c of the manuscript, expressed in mmd-1. The order of magnitude of those evaporations over NA is the same as those over the Mediterranean Sea, up to 0.15 mmh-1. This is better seen in Fig. 5.c. The evaporated moisture is advected with the wind flow, merging with the Atlantic and Mediterranean moisture.

To quantify how much the Mediterranean Sea contributed to the changes of atmospheric moisture at that location, Fig. 5.c shows the contribution from each of the moisture equation terms over the selected volume MED. We can see that between 22-Oct and 26-Oct 1200 UTC there is a positive, homogeneous evaporation from the Sea at a rate of 0.25 mmh-1 that picks up from 26-Oct 1200 UTC to 0.5 mmh-1 by 28-Oct 0000 UTC. The time of the evaporation pick up coincides the occurrence of precipitating convection over the Mediterranean Sea west of Corsica. The intensification of the evaporation is brought by the intensified drag of horizontal winds close to sea surface.  

Figure 5. Analysis of the moisture budget terms. (a) Simulation domains as Fig.1 of the manuscript including the NA and MED areas for calculation of averaged evaporation, precipitation and moisture convergence. (b) Spatial average of hourly IWV variations (dotted grey), Evaporation (blue), Precipitation (red) and Moisture flux Convergence (green) in mmh-1, over investigation area NA between 20-Oct and 23-Oct 0000 UTC. All variables are obtained from the CTRL-7 runs. (c) is as panel (b) but showing the averages over the investigation area MED between 22-Oct and 28-Oct 0000 UTC. (d) is as panel (b) but showing the spatial averages between 21-Oct and 23-Oct 0000 UTC. Mind the changes in the y-axis scaling.

R1C11. The CTRL-7 simulation is validated later (that would be better here or before!)

We will change this accordingly in the next version of the manuscript.

R1C12. no moisture budgets are computed,

See above

R1C13. A validation against MODIS is mentioned but not shown(?!)

We decided not to include this validation against MODIS IWV (version D3, daily product, 1°x1°, onboard Terra and Aqua) since some pixels had missing values. This is due to the inability of MODIS to retrieve IWV when there is cloud cover (Seemann, et al., 2003). Under these lines you can see the validation. Over the eastern Spanish coast and southern France on the 25-Oct-2012 and over the Alps and northern Italy on the 26-Oct-2016 there is no available MODIS data. For the areas with available MODIS data both Terra and Aqua MODIS observations overestimate IWV west of Corsica and Sardinia (25-Oct) and south of Italy (26-Oct) as compared to COSMO-CLM.

Given this issue of too many pixels showing missing values we decided not to include these figures in the manuscript and base our analysis on the model data, radiosondes and GPS.  

Figure 6. Daily averaged IWV measured MODIS onboard Terra (a, b) and Aqua (c, d). The spatial resolution is 1°x1°. The products used are MOD08_D3 and MYD08_D3.

R1C14. Furthermore, I suspect that labels A and B are inverted in the lower panel of Figure 6.b. If I am wrong, it means that I don't understand this panel.

The ellipses A and B try to explain which trajectories are found at which height in the GDAS-HySPLIT simulation. The fastest backward trajectories (those starting at the northwest corner of the island) are denoted with the ellipse A. On the other hand, ellipse B denotes the slower trajectories starting at the southeast corner of the matrix that reach northern Algeria in 24 h. We performed further analyses of these trajectories

using the GDAS-HySPLIT model splitting the matrix of starting points. We saw that the first group of trajectories (ellipse A, reaching the Atlantic Ocean) travel for most of the 24 hours between the levels of 4000 m and 2000 m. On the other hand, the slow trajectories (ellipse B) in the last 12 h of the trajectory descend and travel at a height between 2000 m and 500 m. This is shown in the following graphs, that will be included as supplementary material in the next version of the manuscript. Moreover, a clearer explanation of the figure will be included in order to avoid the misinterpretation of this result.

Figure 7. Lagrangian backward trajectories obtained with the HYSPLIT model, starting on the 26-Oct-2012 1800 UTC (initiation of the event over Corsica) back to for 24 h. Figure 7.a shows the northwest subset of the trajectories shown in Fig.6.d of the manuscript while Fig. 7.b shows the subset corresponding to the trajectories starting at the southeast of the investigation domain. Please not the changes of axis (longitude, latitude and height) shown between panels a and b.

R1C15. L352-353: Why does this "further promote the moisture uptake form the Sea"?

Lifting over the affected area is accompanied by strong low-level convergence close to the surface following the mass-continuity conservation law. Stronger surface winds induce larger evaporation rates over water or wet and vegetated soils (Schneider, et al., 2010; Peixoto & Abraham H. Oort, 1992). In CCLM this process is parameterized through the standard Bulk-Transfer scheme (Louis, 1979) that controls the heat and mass transfer between the surface (land or water) and the atmosphere (Schättler, et al., 2016). In this scheme, the surface water flux affects the atmosphere as lower boundary conditions for the turbulent moisture flux ($F_{(q^v)}^3$) within the turbulent mixing term of the model equations ($M_{(q^v)}^{TD}$). The turbulent moisture flux ($F_{(q^v)}^3$) is proportional to the horizontal wind speed over the surface ($|(v_h)$ ⃗ $|$) following the next equation. $(F_{(q^v)}^3)_{sfc} = -C_q^d |(v_h)$ ⃗ $|(q^v - q_{sfc}^v)$ (1)

Where  is the air density, $C_q^d$ is the bulk-aerodynamical coefficient for turbulent moisture transfer, which is adapted to each surface type (water, soil type, vegetation, etc), $q\_sfc\hat{}v$ is the ground specific humidity and $q\hat{}v$ is the humidity at the lowest model level. Hence, larger moisture flux from the surface (land, sea) to the atmosphere happens with larger horizontal wind speeds. R1C16. L396-398: Looking only at the maximum precipitation value is not really fair because the RG network is rather sparse compared to the scale of the studied phenomenon (and simulated maximum value). At first glance, I would say that CTRL-2.8 is better than NDG-2.8. A proper validation of numerical simulations against RG is needed.

We decided to validate our model results against the HyMEX RG data set given it has undergone several post-processing procedures (four revised versions of the product, https://mistrals.sedoo.fr/?editDatsId=904&datsId=904&project_name=HyMeX) and enjoys a large quality. We performed another validation against the CMORPH product, which has a spatial scale (8 km), similar to the model resolutions (7 km and 2.8 km) but found large discrepancies between CMORPH and RG for this case study. Therefore, we dismissed the analysis against CMORPH.

We acknowledge that a point-to-point comparison against RG entails difficulties, such as the sparse distribution over the island, as pointed out by the reviewer, and the double-penalty problem due to missed location of precipitation maxima (Wernli, et al., 2008; Gilleland, et al., 2009). This is why, in the comparison against RG (Tables 1 and 3) we perform a point-to-point comparison interpolating to the location of the stations but also analyse the maximum and standard deviation of the precipitation distributions intensities over the island. Since RG are usually employed as reference data for precipitation validation (Habib, et al., 2012; Jiang, et al., 2018) we decided to use this data set for the validation.

We show here the validation metrics similarly to Table 1 for the seasonal simulations (initialized on 01-Sep-2012). For the calculation of the metrics, hourly and accumulated precipitation values between 26-Oct 1300 UTC and 27-Oct 1700 UTC i.e. the period of precipitation over CO. We confirm the comment of the reviewer for the 2.8 km

since CTRL-2.8 outperforms NDG-2.8 in the simulation of the standard deviations of precipitation (both hourly and accumulated), accumulated precipitation and maximum. Only, the hourly RMSE is improved in NDG-2.8.

Regarding the 7 km simulations, applying the GPS-ZTD nudging is beneficial in that it improves the RMSE, and the standard deviation and maximum value of the accumulated precipitation values. This is as described in the manuscript.

These results can be explained from the precipitation reduction of the NDG-7 and NDG-2.8 simulations. CTRL-7 was overestimating excessively precipitation, whereas CTRL-2.8 showed a good representation of the maximum and accumulated amount. After a drying, only the scores of the 7 km resolution were improved, but not those of the 2.8 km.

We will adapt our conclusions and analysis in the manuscript to suit these findings. Besides, we will include Table 3 in Section 4.1 of the manuscript.

Table 3. As, Table 1 for the runs initialized on the 1-Sep-2012. The validation is done against hourly or accumulated precipitation values between 26-Oct 1300 UTC and 27-Oct 1500 UTC i.e. the period of precipitation over CO.

R1C17. L400: The authors do not really assess the accuracy of model moisture outputs: since most of the radiosonde locations are near GPS receivers, they rather assess the accuracy of GPS IWV retrievals. This most certainly explains why NDG-7 and NDG-2.8 results are so close to each other. To really assess the accuracy of model moisture outputs, free runs should be evaluated.

Only two sets of simulations can be validated against radiosondes. On the one hand, the free runs (CTRL), where no observation nudging is performed and hence are runs constrained only by the forcing data. On the other hand, the NDG runs where IWV is corrected every 10 minutes. We are assessing the accuracy of model moisture output for these two different sets of simulations.

Regarding the comment on the proximity between the GPS stations and the radiosondes. We must add that CCLM redistributes the nudged GPS information in the vertical profile specific humidity. Hence CCLM constructs a profile where errors at some levels are introduced. This is exactly what we wanted to quantify in Section 4.2 and Figures 8 and 9.

R1C18. L411: Figure 8.a shows the IWV over the CO domain, not Corsica. This is important, because the most noticeable differences among the nudging simulations may be over the sea since IWV mainly comes from GPS receivers on ground. The authors should adapt the rest of their interpretation of Figure 8.a accordingly.

Yes, thanks for this remark. Indeed, the spatial averages shown in Fig.8.a are obtained from the area CO and not Corsica. The text has been changed accordingly in the manuscript. In now reads: Figure 8.a shows the spatially averaged temporal evolution of IWV over CO. The hours prior to precipitation initiation (26-Oct 1300 UTC) were characterized by an IWV pick up starting at 26-Oct 0000 UTC. All simulations show this, albeit the IWV amount over CO for NDG-7 and NDG-2.8 was 5 mm higher than for CTRL-7 and CTRL-2.8. This was due to represented precipitation over the island until the night of 24-Oct in the NDG runs, hence inducing a much wetter boundary layer (not shown).

R1C19. L423: Where is it evidenced that the humidity reduction takes place below 500 hPa?

How the GPS-ZTD nudging affects the vertical distribution of humidity is introduced in Section 4.2, in Figs. 8.b and Fig. 9. This is done on the one hand through box and whiskers plots over the investigation domain CO (Fig. 8.b) and on the other hand in the validation of the CCLM atmospheric moisture distribution against radiosondes (Fig 9). The sentence, as pointed out by the reviewer, could be misleading since no reference to the figures is given. We have corrected this in the manuscript. Furthermore, Fig. 8 shows how between 26-Oct-2012 1300 UTC and 27-Oct-2012 1500 UTC the specific
humidity median, quartile and extreme values at 500 hPa are lower in NDG as compared to CTRL. We will revise our manuscript to read "The humidity reduction between 26-Oct 1600 UTC and 27-Oct 0600 UTC takes mostly place at 500 hPa down to the 950 hPa level".

R1C20. L429-447: A validation against radiosondes is presented. Which radiosonde profile has an impact on the event under consideration? They are all located either east, north, or south of Corsica, while the authors showed that moisture comes from the (south-)west.

The validation against radiosondes of Section 4.2 had for objective showing how good was the performance of the model in representing the profile of specific humidity. We selected all operational stations available during the two days of convective activity in the Mediterranean contained within the simulation domains of both resolutions. Hence the seven stations shown in Fig. 1 of the manuscript. Even if the stations are located only east of the study region on the Italian Peninsula, they are still valuable to assess the performance of the model.

R1C21. L466-467: Where is a "decrease of humidity close to ground" shown?

Thanks for this remark. In the analysis of box and whisker plots for 2m specific humidity we saw a reduction of ca. 1 gkg-1 and 0.5 gkg-1 in the NDG-7 and NDG-2.8 respectively as compared to their CTRL counterparts. This was shown for the period between 26Oct 1300 UTC and 27-Oct 1500 UTC over Corsica. We did not include this graph in our manuscript in order to make the text more concise, but we will include it as a subpanel of Fig. 8.b in the next version of our manuscript. Figure 8.b then will show as it follows

Figure 8. Spatially averaged IWV for all simulations during the event. The area for averaging is shown in Fig. 1 (CO) and the model output has been upscaled to a common coarser grid. The period shown is 26-Oct 0000 UTC to 28-Oct 0000 UTC. (b) Box and whiskers plots showing the median, the percentiles 25 and 75 and the extreme

values of IWV and specific humidity at 500 hPa, 700 hPa,950 hPa and at 2m height. All box and whiskers are obtained from the distribution of values for the shown quantities between the 26-Oct 1300 UTC and 27-Oct 1500 UTC over the study region CO.

R1C22. L469-471: A simulation nudging IWV at 2.8 km and forced by NDG-7 would be useful here.

As explained in Section 1, we decided to force the NDG-2.8 runs with CTRL-7 to allow for comparison against CTRL-2.8. We believe this is the best means to assess the direct impact of the GPS-ZTD nudging in this resolution, avoiding any potential influences coming from the forcing data.

R1C23. L487: Why is the effect of the low-pressure system change exclusively seen in the 7 km simulations? Isn't the wind changed in the 2.8 km simulations, too?

We show only the results in the 7 km resolution since these runs cover the whole western Mediterranean area, as opposite to the 2.8 km runs. The 2.8 km simulation domain only covers the Italian Peninsula and the islands in order to save computational resources. The effect described in the manuscript (increase of PMSL at the centre of the low-pressure system in southern France) is only seen in the NDG-7 simulation since NDG-2.8 does not cover southern France. The following figure shows the PMSL changes in the NDG-2.8 compared to CTRL-2.8 and in the winds at 950 hPa (analogously to Fig. 12 in the manuscript for the 7 km runs).

Figure 9. Spatial distribution of the differences in Pressure at the Mean Sea Level (PMSL) between NDG-2.8 and CTRL-2.8 on the 26-Oct 2300 UTC. Horizontal winds at 950 hPa are represented by black (CTRL-2.8) and green (NDG-2.8) arrows.

Fig. 9 shows that there are some areas (Adriatic Sea, the Balkans, southern Alps) with lower PMSL in the NDG-2.8 than CTRL-2.8 in extent of -2 hPa. These differences are weaker than the differences observed in the 7 km runs (up to 10 hPa) and are of opposite sign. Fig.9 clearly shows that the NDG-2.8 run does not show the same effect

as NDG-7. The weakening of the low-pressure system over southern is not simulated in the NDG-2.8. There are however changes in the wind direction and intensity, mainly offshore, ahead of the western Italian and the Balkan coasts. However, these changes in wind direction and speed do not correspond to the changes observed in the 7 km runs since they do not show a clockwise veering of the direction centered in the southern French region. We will also include this graph in the supplementary material to support the analysis of the PMSL variations in Section 4.4

R1C24. L568-569: The high-temporal resolution of GPS-ZTD observations has not been shown to facilitate a better representation of the water vapour variability and a better regulation of the accumulated precipitation. To do this, different temporal resolutions should have been used. In the general comments part, we have addressed how only little differences arise from using different nudging frequencies for spatially averaged precipitation in the 7 km for this case study over CO. These findings will be presented in the new version of the manuscript in Section 4.1 (Nudging effects on Precipitation).

3-TECHNICAL CORRECTIONS We have accepted all corrections and comments of this section in the revised version of the manuscript. Where needed a short clarification is added here.

R1C25. L217: Is it useful to describe the vertical interpolation in the nudging scheme when it comes to assimilate GPS data?

We have adapted these lines in the manuscript to give this information in a clearer way. How the information is spread in the vertical direction is relevant since each GPS observation is used to construct a specific humidity profile that is treated as such in the nudging scheme. Hence, what is used for the nudging is this constructed profile based on the issued GPS-ZTD value. This sentence has been rewritten as: "The vertical interpolation of the observed data is performed assuming a Gaussian decay in height differences. The vertical interpolation is also applied in the case of GPS-ZTD

nudging since a profile of specific humidity is constructed from the derived GPS-IWV value. This constructed profile shall be treated by the nudging scheme as an upper-air measurement in the remainder of the process."

R1C26. L230-231: It is unclear. What is the "iterative process"? If Eq 2 is used, it is not an iterative process, is it?

Yes, it is an iterative process that its repeated until a sufficiently low error is reached or after 20 iterations. We acknowledge that Eq. (2) in the manuscript was not written as an iterative formula and we have corrected this. This paragraph now reads.

"The observations are assigned to a grid point in the model space, provided the altitude difference of the GPS station and model surface lays within the range -150 to 600 m to allow for extrapolation and interpolation, respectively and are converted to a specific humidity profile ($q\_v\hat{}mod$). This is needed given IWV is not a model prognostic variable as opposite to specific humidity. The profile is constructed by means of an iterative process that scales the observed IWV (ãĂŰIWVãĂŮˆobs) with the modelled one (ãĂŰIWVãĂŮˆmod) until a sufficiently low error is reached or up to 20 iterations. Eq. (2) describes the iterative formula. The first profile ($q\_(v\_i)\hat{}mod$) used as the first guess for the iterative process, is the modelled specific humidity profile. Hence, the profile used for nudging depends on the vertical humidity distribution simulated by the model at the beginning of the nudging time-window. "

R1C27. L241: It is stated that the 7 km runs are forced by ECMWF analyses. How often? Every 6 h. The forcing data has a temporal resolution of 6 h. We have included this information in the newest version of the manuscript.

R1C28. L244: The terms "large precipitation reductions" is unclear. The authors must specify which experiences they are referring to.

This sentence has been rephrased to express the information in a clearer way. In it, we were referring to a large reduction of maximum precipitation that was induced by

the GPS-ZTD nudging. We have added more details so that this sentence is not out of context. The sentence now reads.

"Within the 80-day period of simulation, there were several events, which were largely affected by the GPS-ZTD nudging. IOP16, the case study of this paper, is one of them which is especially interesting given the large reduction of maximum precipitation (-20 %) induced by the GPS-ZTD nudging over the investigation area of Corsica in the course of 26 h. IOP16 is also suitable to assess the benefit of atmospheric moisture corrections with GPS-ZTD nudging given the important role of the local orographic and instability factors in triggering and maintaining convection rather than the large-scale upper level forcing."

R1C29. L298: What is an "offshore-size" convective system?

Thy is a typo. The sentence should read "Between 26-Oct 1900 UTC and 27-Oct 0100 UTC, offshore convective systems arrive at the island (see Fig.3.a)". We have corrected the text accordingly.

R1C30. L447: What is an "accuracy rate"?

The corrected sentence is: "The 2.8 km simulation was initially more accurate, but the nudging brings both to similar accuracy values".

R1C31. L454: What does "height-surface" mean? Do the authors simply mean "vertical"?

Yes, it was meant vertical cross-section. Thank you. It has been corrected in the manuscript.

R1C32. L455: What does "vertical-horizontal direction" mean?

The sentence has been changed to: "Figure 10 shows the vertical cross-sections of Equivalent Potential Temperature ($\theta\_e$), specific humidity and the wind along the direction of the mean horizontal wind (purple transect in Fig. 1) over the island at 26-Oct

1700 UTC."

4-REFERENCES Benjamin, S. G. et al., 2010. Relative Short-Range Forecast Impact from Aircraft, Profiler, Radiosonde, VAD, GPS-PW, METAR, and Mesonet Observations via the RUC Hourly Assimilation Cycle. Monthly Weather Review, 4, Band 138, p. 1319–1343. Borderies, M. et al., 2019. Assimilation of wind data from airborne Doppler cloud-profiling radar in a kilometre-scale NWP system. Natural Hazards and Earth System Sciences, 4, Band 19, p. 821–835. Gilleland, E. et al., 2009. Intercomparison of Spatial Forecast Verification Methods. Weather and Forecasting, 10, Band 24, p. 1416–1430. Habib, E., Haile, A. T., Tian, Y. & Joyce, R. J., 2012. Evaluation of the High-Resolution CMORPH Satellite Rainfall Product Using Dense Rain Gauge Observations and Radar-Based Estimates. Journal of Hydrometeorology, 12, Band 13, p. 1784–1798. Jiang, Q. et al., 2018. Accuracy Evaluation of Two High-Resolution Satellite-Based Rainfall Products: TRMM 3B42V7 and CMORPH in Shanghai. Water, 1, Band 10, p. 40. Keil, C., Heinlein, F. & Craig, G. C., 2013. The convective adjustment time-scale as indicator of predictability of convective precipitation. Quarterly Journal of the Royal Meteorological Society, 5, Band 140, p. 480–490. Lamb, P. J., Portis, D. H. & Zangvil, A., 2012. Investigation of Large-Scale Atmospheric Moisture Budget and Land Surface Interactions over U.S. Southern Great Plains including for CLASIC (June 2007). Journal of Hydrometeorology, 12, Band 13, p. 1719–1738. Louis, J.-F., 1979. A parametric model of vertical eddy fluxes in the atmosphere. Boundary-Layer Meteorology, Band 17, pp. 187-202. Mahfouf, J.-F., Ahmed, F., Moll, P. & Teferle, F. N., 2015. Assimilation of zenith total delays in the AROME France convective scale model: a recent assessment. Tellus A: Dynamic Meteorology and Oceanography, 2, Band 67, p. 26106. Peixoto, J. P. & Abraham H. Oort, 1992. Physics of Climate. s.l.:American Inst. of Physics. Sahlaoui, Z., Mordane, S., Wattrelot, E. & Mahfouf, J.-F., 2019. Improving heavy rainfall forecasts by assimilating surface precipitation in the convective scale model AROME: A case study of the Mediterranean event of November 4, 2017. Meteorological Applications, 12.Band 27. Schättler, U., Doms, G. & Schraff, C., 2016. A Description of the Nonhydrostatic Regional COSMO-Model Part

VII : User's Guide, DeutscherWetterdienst, P.O. Box 100465, 63004 Offenbach, Germany: s.n. Schneider, M. et al., 2010. Continuous quality assessment of atmospheric water vapour measurement techniques: FTIR, Cimel, MFRSR, GPS, and Vaisala RS92. Atmospheric Measurement Techniques, 3, Band 3, p. 323–338. Seemann, S. W., Li, J., Menzel, W. P. & Gumley, L. E., 2003. Operational Retrieval of Atmospheric Temperature, Moisture, and Ozone from MODIS Infrared Radiances. Journal of Applied Meteorology, 8, Band 42, p. 1072–1091. Wernli, H., Paulat, M., Hagen, M. & Frei, C., 2008. SAL—A Novel Quality Measure for the Verification of Quantitative Precipitation Forecasts. Monthly Weather Review, 11, Band 136, p. 4470–4487.

Please also note the supplement to this comment:
https://www.nat-hazards-earth-syst-sci-discuss.net/nhess-2019-319/nhess-2019-319-AC1-supplement.pdf

---

## Author Comment (AC2) · 22 Mar 2020

Answers to reviewer 2 (Paper-IOP16-NHESS)

GENERAL COMMENT

We would like to thank the anonymous reviewer for the valuable comments, suggestions and questions. We have considered all minor and major comments. We believe that the quality of the manuscript has increased, thanks to the comments of the reviewer and acknowledge how answering the raised questions was crucial for a better

exposition of our key messages.

R2C1. The impact of evapotranspiration over North-Africa for this event is shown, even if not quantified.

We give a description on how much moisture evaporates from North-Africa between lines 344 and 361 using our model simulations and the GLEAM product (combination of different satellite measurements). We give values on how much moisture evaporates per day in the days prior to the event at Corsica. For example, on line 359 we state "Suite to the precipitation event, daily evapotranspiration over the area reached spatial averages of 1.4 mm as shown by GLEAM, lasting for seven days, well above the mean evapotranspiration values during this season (0.5 mm). Albeit differences in the 360 magnitude of evaporation, COSMO-CLM well captures this period of anomalous evapotranspiration".

How the moisture is further transported north towards Corsica is explained in the following paragraphs of Section 3.3.

A quantification of the contribution of evapotranspiration over north Africa in relation to the advected moisture is presented in our answers to the reviewer.

MAJOR POINTS

R2C1.a) The authors must clarify that this is a diagnostic study and not prognostic. The impact of assimilating GPS-ZTD is quantified by comparing two simulations: the first doesn't assimilate GPS-ZTD, while the second assimilates GPS-ZTD continuously. While this is an important comparison, it must be clarified that the paper doesn't assess the role of GPS-ZTD in a prognostic approach for the case study.

We agree with the reviewer that this point could be better clarified to the reader, for that reason we have included this information in several relevant places in the manuscript:

In the Abstract "In this study, we use a diagnostic approach to assess the sensitivity of precipitating convection and underlying mechanisms during a heavy precipitation event
(HyMeX intensive observation period 16) to corrections of the atmospheric moisture spatio-temporal distribution."

In Section 2.2.2 of the Methods. The GPS-ZTD Nudging Sensitivity Experiments "The Nudging scheme is used to assimilate GPS-ZTD data to assess the sensitivity of heavy precipitating convection to corrections of the spatio-temporal distribution of atmospheric moisture. We use a diagnostic approach as opposite to commonly use data-denial experiments."

R2C1.b) Also the importance of sub-hourly data assimilation is not shown. To do that a comparison between two simulations one assimilating GPS-ZTD on an hourly basis and the other one assimilating GPS-ZTD every 10 minutes (as in the paper) should be performed. However, I understand that this requires adding new simulations, which can be avoided deleting the sentences where the importance of sub-hourly assimilation is emphasised.

We agree that this is a very interesting question that needs to be addressed in our paper.

We have performed supplementary simulations, initialized on the 20-Oct-2012 with no GPS-ZTD nudging (CTRL), 10-min frequency nudging (NDG-7 and NDG-2.8) and 1 h frequency nudging (NDG-7-1h and NDG-2.8-1h) where the temporal resolution of the GPS data is of 1 h (NDG-7-1h, NDG-2.8-1h). The only difference between the 1h nudging and the 10min nudging simulations is the frequency of assimilation. All other settings are the same. We have performed these experiments starting the simulations on the 20-Oct-2012 at 0000 UTC, to reduce the computational costs of running the complete season.

In the following we show graphs and analysis for the comparison between the runs with nudging frequency of 10 min vs. 1h.

Figure 1. Spatially averaged IWV (a, c) and Precipitation (b, d) for the 7 km simulations

(a, b) and the 2.8 km simulations (c, d) during the event. The area for averaging is Corsica. The model output has been upscaled to a common coarser grid of 8 km. The period shown is 26-Oct 0000 UTC to 28-Oct 0000 UTC. The comparison is between the runs using a temporal nudging frequency of 10 min (NDG-7, NDG-2.8) against nudging at a temporal frequency of 1h (NDG-7-1h, NDG-2.8-1h).

The results show no differences on the temporal evolution of IWV. This holds for 7 km and 2.8 km. This can be explained by the fact that we calculate the spatially averaged IWV at sharp hours (i.e. 0000 UTC, 0100 UUTC, 0200 UTC, etc.), precisely is at those times when the GPS-ZTD data is assimilated in the NDG-7-1h and the NDG-2.8-1h runs.

For precipitation, there is a slight impact for the 7 km runs, but not for the 2.8 km. The NDG-7-1h simulation shows somewhat larger precipitation than NDG-7 at 2000 UTC on the 26-Oct (Fig. 1b) corresponding to an increase from 30 mm to 50 mm at the western shore of the island Fig.2.b and Fig. 2.c

Figure 2. COSMO-CLM accumulated precipitation over Corsica between 26-Oct 1300 UTC and 27-Oct 1500 UTC i.e. during the period of precipitation over the island and RG.

To delve further into which aspects of precipitation representation have been improved we present in Table 1 further validation metrics using the Rain Gauges (RG) as reference.

Table 1 shows the RMSE of the anomalies of hourly precipitation rates (first column), the differences (OBS-MOD) of the standard deviations of hourly precipitation (second column) and the spatially averaged differences of accumulated precipitation during the whole event, i.e. between 26-Oct 1300 UTC and 27-Oct 1500 UTC (third column). The last three metrics are obtained after interpolating the COSMO-CLM precipitation values to the location of the RG stations. On the contrary, columns four and five of Table 1, show differences of the standard deviation and maximum value of precipitation

for COSMO-CLM over land without interpolation. That means, we have obtained all 27h-accumualeted precipitation values simulated by COSMO-CLM over land and have calculated the standard deviation and the maximum. We have done the same for all RG measurements and the differences are shown. We do this to avoid double-penalty problems due to a possible misrepresentation of the maxima location (Wernli, et al., 2008; Gilleland, et al., 2009). The formulas used are included in Table 2.

Table 1. Metrics of the precipitation validation against RG. The model precipitation has been interpolated to the location of the RG for the first three columns and all precipitation values simulated by COSMO-CLM over the island of Corsica are used in the last two columns. This is done to avoid double-penalty problems due to a shifting of the precipitation maxima. N is the number of RG stations and M the total number of grid points. The units are mm.

Table 2. Precipitation validation metrics.

Overall, we see that nudging GPS-ZTD data is beneficial for the 7 km grid with little difference between nudging with 1h frequency or 10 min. If any, we see a slight advantage in nudging GPS-ZTD data with 10 min for the representation of the hourly standard deviation rates. The same holds for the 2.8 km, assimilating with a 10min frequency shows very weak differences with respect to assimilating with 1h frequency.

These analyses will be included as an additional subsection in section 4.1 (Nudging effects on Precipitation).

R2C2) In the section 3.3 emphasis is given to the transport of humidity from North Africa for the event. It would be interesting to give a comparison between this source of moisture and that coming from the western Mediterranean Sea to define better this contribution. We agree with the reviewer that a quantification of the different terms of Evaporation and moisture flux over the investigation area NA and the Mediterranean would be most interesting. We have obtained the terms, described in Lamb et al. (2012) over the investigation areas NA (North Africa) and MED (Mediterranean Sea)

for this purpose, see Fig. 3.a.

The calculation of these terms entails simplifications for example, of the turbulent and microphysical processes that introduce relevant uncertainties. Hence, what we provide here is an estimation of the different contributions.

$\Delta$IWV=E+(-P)+MFC (1)

All terms of Eq. 1 are expressed in mmh-1, where positive signs of the Evaporation (E) and Integrated Moisture Flux Convergence (MFC) imply an increase of Integrated Water Vapour variations ($\Delta$IWV>0) within the NA and MED volumes. On the contrary, if precipitation and water vapor divergence occur (MFC<0) IWV decreases ($\Delta$IWV<0). The volumes cover the areas in Fig. 3.a where the integrations of IWV and MFC are performed from the first to the last model levels.

Fig. 3.b shows similar information to Fig.6.c of the manuscript. Intense precipitation occurs over NA on the 20-Oct-2012 with the subsequent decrease of IWV, and intense evaporation over the area on the 21-Oct and 22-Oct at midday. This is the moment when solar radiation is strongest, hence evaporation is intensified over this wet soil. Please note the change in the axis scales between the different panels of this figure and those of Fig.6.c of the manuscript, expressed in mmd-1. The order of magnitude of those evaporations over NA is the same as those over the Mediterranean Sea, up to 0.15 mmh-1. This is better seen in Fig. 3.c. The evaporated moisture is advected with the wind flow, merging with the Atlantic and Mediterranean moisture.

To quantify how much the Mediterranean Sea contributed to the changes of atmospheric moisture at that location, Fig. 3.c shows the contribution from each of the moisture equation terms over the selected volume MED. We can see that between 22-Oct and 26-Oct 1200 UTC there is a positive, homogeneous evaporation from the Sea at a rate of 0.25 mmh-1 that picks up from 26-Oct 1200 UTC to 0.5 mmh-1 by 28-Oct 0000 UTC. The time of the evaporation pick up coincides the occurrence of precipitating convection over the Mediterranean Sea west of Corsica. The intensification of the

evaporation is brought by the intensified drag of horizontal winds close to sea surface.

Figure 3. Analysis of the moisture budget terms. (a) Simulation domains as Fig.1 of the
manuscript including the NA and MED areas for calculation of averaged evaporation,
precipitation and moisture convergence. (b) Spatial average of hourly IWV variations
(dotted grey), Evaporation (blue), Precipitation (red) and Moisture flux Convergence
(green) in mmh-1, over investigation area NA between 20-Oct and 23-Oct 0000 UTC.
All variables are obtained from the CTRL-7 runs. (c) is as panel (b) but showing the
averages over the investigation area MED between 22-Oct and 28-Oct 0000 UTC. (d)
is as panel (b) but showing the spatial averages between 21-Oct and 23-Oct 0000 UTC.
Mind the changes in the y-axis scaling.

R2C3.a) Considering the nudging scheme there is no information on the parameters
of the Second order autoregressive function. How they are determined?

We have added a short clarification in the revised version of the manuscript. In Eq.
(1.b), $\Delta r$ stands for the distance between the observation location and the target grid
point. The other parameter, s, is the correlation scale that is defined in tables with a
pressure level dependency for the different variables (Schraff & Hess, 2012). For ex-
ample, for temperature ($s\_T$) and humidity ($s\_q$) these values are: Table 3. Correlation
scale (s) for temperature (T) and specific humidity (q) at different pressure heights

The correlation scale, thus, is larger in the stratosphere than in the troposphere and
lowest levels. As an example, at 500 hPa, the weight for the horizontal spreading is
halved ($w\_xy=0.5$) at a horizontal distance of about 135 km from the station location.

The revised explanation in the manuscript reads "The weight for spreading in the hor-
izontal direction $w\_xy$, is a second-order autoregressive function dependent on the
distance between the observation and the target point ($\Delta r$) and the correlation scale
(s), see Eq. (1.b). The correlation scale is dependent on the pressure level and is
largest for the stratosphere (100 km) and shortest for the PBL (58 km). This implies

that the horizontal weight is halved (w_xy=0.5) at a distance of 135 km at the 500 hPa level"

R2C3.b) Line 217 has a comment on the vertical adjustment that doesn't apply to the specific case.

We have adapted these lines in the manuscript to give this information in a clearer way. How the assimilated observation is adjusted in the vertical direction is indeed relevant since each GPS observation is used to construct a specific humidity profile that is treated as such in the nudging scheme. Hence, what is used for nudging is this "constructed" profile based on the issued GPS-ZTD value. This sentence has been rewritten as: "The vertical interpolation of the observed data is performed assuming a Gaussian decay in height differences. The vertical interpolation is also applied in the case of GPS-ZTD nudging since a profile of specific humidity is constructed from the derived GPS-IWV value. This constructed profile shall be treated by the nudging scheme as an upper-air measurement in the remainder of the process."

R2C3.c) It also unclear how the qv profile is constructed iteratively (Lines 230-232). Do you mean that it is modified by nudging until a difference is attained or something different?

For a certain GPS-ZTD observation at time t and location x ⃗ the ZTD value is converted to IWV. This is described in the paper following Schraff and Hess (2012). Given IWV, as opposite to specific humidity, is not a prognostic variable of the model, the IWV observation has to be transformed into a profile of specific humidity. This is done for each single observation by means of an iterative process. To express this process in a more understandable way the whole paragraph has been rewritten.

"The observations are assigned to a grid point in the model space, provided the altitude difference of the GPS station and model surface lays within the range -150 to 600 m to allow for extrapolation and interpolation, respectively and are converted to a specific humidity profile ($q_v^{mod}$). This is needed given IWV is not a model

prognostic variable as opposite to specific humidity. The profile is constructed by means of an iterative process that scales the observed IWV (ãĂŰIWVãĂŮˆobs) with the modelled one (ãĂŰIWVãĂŮˆmod) until a sufficiently low error is reached or up to 20 iterations. Eq. (2) describes the iterative formula. The first profile (q_(v_i)ˆmod) used as the first guess for the iterative process, is the modelled specific humidity profile. Hence, the profile used for nudging depends on the vertical humidity distribution simulated by the model at the beginning of the nudging time-window. " q_(v_(i+1))ˆmod=q_(v_i)ˆmod·ãĂŰIWVãĂŮˆobs/(ãĂŰIWVãĂŮ_iˆmod )

MINOR POINTS

All minor corrections mentioned by the reviewer have been accepted and included in the revised version of the manuscript. When needed, a short clarification is included here.

R2C4. Line 33: During heavy precipitation events, rain rates can be much higher than 20 mm/h.

Yes, thanks for the comment. As reported for example in Röhner, et al., (2016) and Ducrocq, et al., (2014), some extreme past events have shown precipitation totals reaching 500 mm in 6 h to 12 h in the Mediterranean area. This is however very exceptional cases. We have adapted the sentence in the manuscript, now it reads.

"During these events, daily accumulated precipitation over 150 mm is not rare and precipitation rates can be well over 20 mm h-1 (Röhner, et al., 2016; Ducrocq, et al., 2014)" R2C5. Line 244: Check the sentence "given the large precipitation reduction". Do you mean when you assimilate GPS-ZTD?

Yes. This sentence has been rephrased to express the information in a clearer way. In it we were referring to a large reduction of maximum precipitation that was induced by the GPS-ZTD nudging. We have added more details so that this sentence is not out of context. The sentence now reads.

"Within the 80-day period of simulation, there were several events, which were largely affected by the GPS-ZTD nudging. IOP16, the case study of this paper, is one of them which is especially interesting given the large reduction of maximum precipitation (-20 %) induced by the GPS-ZTD nudging over the investigation area of Corsica in the course of 26 h. IOP16 is also suitable to assess the benefit of atmospheric moisture corrections with GPS-ZTD nudging given the important role of the local orographic and instability factors in triggering and maintaining convection rather than the large-scale upper level forcing."

R2C6. Line 248: I suggest giving more details about the Agreement Index (AI).

The formula for AI calculation has been included in the revised version of the manuscript. Together with the Agreement Index, more validation metrics are used for precipitation. They are contained in Table 2.

R2C7. Line 577-578: This sentence is rather unclear. It is important to note that, in general, the adjustment introduced by GPS-ZTD could be a function of the height if variational approaches are considered, through the background error matrix. Indeed, generalizing to other assimilation schemes is not possible from our study. This conclusion relating to the difficulties of the nudging scheme to correctly improve the vertical distribution of humidity when using GPS should be constrained to this type of observation and assimilation method, exclusively. In the sentence mentioned by the reviewer it is explicitly said "GPS-ZTD nudging". We agree, nevertheless, that it would be very interesting extending these experiments to other assimilation schemes. This was one of our initiatives in the project that could sadly not be finished due to missing forcing data from the ICON ensemble for the year 2012.

REFERENCES Ducrocq, V. et al., 2014. HyMeX-SOP1: The Field Campaign Dedicated to Heavy Precipitation and Flash Flooding in the Northwestern Mediterranean. Bulletin of the American Meteorological Society, 7, Band 95, p. 1083–1100. Gilleland, E. et al., 2009. Intercomparison of Spatial Forecast Verification Methods. Weather

and Forecasting, 10, Band 24, p. 1416–1430. Lamb, P. J., Portis, D. H. & Zangvil, A., 2012. Investigation of Large-Scale Atmospheric Moisture Budget and Land Surface Interactions over U.S. Southern Great Plains including for CLASIC (June 2007). Journal of Hydrometeorology, 12, Band 13, p. 1719–1738. Röhner, L., Nerding, K.-U. & Corsmeier, U., 2016. Diagnostic study of a HyMeX heavy precipitation event over Spain by investigation of moisture trajectories. Quarterly Journal of the Royal Meteorological Society, 6, Band 142, p. 287–297. Schraff, C. & Hess, R., 2012. A Description of the Nonhydrostatic Regional COSMO-Model Part III: Data Assimilation, DeutscherWetterdienst, P.O. Box 100465, 63004 Offenbach, Germany: s.n. Wernli, H., Paulat, M., Hagen, M. & Frei, C., 2008. SAL—A Novel Quality Measure for the Verification of Quantitative Precipitation Forecasts. Monthly Weather Review, 11, Band 136, p. 4470–4487.

Please also note the supplement to this comment:
https://www.nat-hazards-earth-syst-sci-discuss.net/nhess-2019-319/nhess-2019-319-AC2-supplement.pdf

---

## Author Comment (AC3) · 25 Mar 2020

The comment was uploaded in the form of a supplement: https://www.nat-hazards-earth-syst-sci-discuss.net/nhess-2019-319/nhess-2019-319-AC3-supplement.zip

---

## Referee Report (RR1)

[referee-annotated manuscript omitted]

---

## Author Response (AR2)

**Answers to reviewer 1 (Second revision)**

We would like to thank the anonymous reviewer for the valuable comments and corrections. In the following, we include a detailed answer to the questions/requests. The corresponding changes have been included in the new version of the manuscript and can be followed in the "track_changes" document.

1 - SPECIFIC COMMENTS

*L200: What is k_obs? Is it simply k?*

Yes, k_obs is simply k, which is an index running over all observation types. We corrected the explanation in the manuscript to offer a clearer description. To further explain the meaning of "k" in the nudging equation, part 3 of the COSMO manual says in page 10 (Schraff and Hess, 2012; http://www.cosmo-model.org/content/model/documentation/core/default.htm#p3). "$\varphi_k^{obs}$ [is] the value of the k-th observation influencing the grid point x at time t".

*L202: I don't understand how W_K and w_xy, w_z, and w_t are connected.*

For a single observation type in the assimilation procedure, as is the case of our study, $W_k$ is expressed as $w_k$ and depends on the temporal and spatial weights ($w_{xy}, w_z$), as well as the quality and representativeness of the observation ($\varepsilon_k$) by means of the relation $w_k = w_{xy} \cdot w_z \cdot w_t \cdot \epsilon_k$.

If more than one observation types are assimilated, the observation weights for each of them are combined by the formula $W_k = w_k / \sum_j w_j \cdot w_k$ where $w_j$ is an additional weight for each observation type. This is done to ensure that the contribution of the assimilation increments do not dominate over the solutions from the physics and dynamics.

This information has been included in the new version of the manuscript.

*L207: I don't understand why vertical interpolations are needed. I understood (perhaps wrongly) that IWV has no vertical coordinate and that the constructed specific humidity profiles are given on model levels.*

Thanks for this comment. It is as you say. GPS derived specific humidity profiles need not be interpolated from height levels to model levels, as they are already constructed on them. This has been corrected in the manuscript.

*L221-227: The authors should explain why this procedure does not converge in one step*

Several iterations are needed to scale the specific humidity at each level to match the observed IWV since sometimes specific humidity surpasses saturation at one or several levels. When this happens specific humidity at those pressure levels is truncated to saturation, altering in turn the IWV value, leaving it sometimes far away from the observed one. Hence the need to implement the scaling in an iterative routine. A short explanation has been added to the manuscript.

*L325-328: The validation against MODIS is interesting. Can the authors explain why TERRA and AQUA products differ? The visual comparison would be much easier to appreciate if MODIS and COSMO color scales were the same.*

The TERRA and AQUA MODIS products differ because they are obtained by two different MODIS instruments aboard the Terra and Aqua satellites. Terra passes from north to south across the equator in the morning and

Aqua passes from south to north over the equator in the afternoon. Hence the shown product, the daily IWV with 1°X1° resolution, is obtained as an average of all available passages of each satellite over the pixels within the shown investigation area. Given the TERRA and AQUA swaths pass over the investigation areas at different times of the day, the products give a different estimation of the daily mean IWV.

The following Figure (Fig.S.1 in the annex) has been put to the same colour scale as Fig.4 in the manuscript.

[Figure]

**Figure 1. Daily averaged IWV measured MODIS onboard Terra (a, b) and Aqua (c, d). The spatial resolution is 1°x1°. The products used are MOD08_D3 and MYD08_D3.**

*L364-365: How confident are the authors that GDAS is as accurate as COSMO-CLM?*

The Global Data Assimilation System provides model analysis based on the Global Forecasting System atmospheric model, assimilating observations by means of a 3D Var until 2016 (Hoover et al., 2017) and a 4D Ensemble-Variational afterwards (https://www.emc.ncep.noaa.gov/gmb/STATS/html/model_changes.html). The observation types assimilated are, among others surface stations, wind profilers, aircrafts, buoys, radar, and satellite observations (IASI, ATMS, ASCAT, etc.;
https://www.emc.ncep.noaa.gov/gmb/STATS/html/model_changes.html).
GDAS is a model from the National Center for Environmental Prediction (NCEP) that has been evaluated in early and recent publications (Dey and Morone, 1984; Andrei and Chen, 2008; Kleist et al., 2009) and is widely used as testbench for data assimilation experiments of new observation types. For example, the assimilation of Rapidscat surface wind retrievals (Liu et al., 2018) or stratospheric balloons (Lukens, 2019, AGU Abstract).

Additionally, we have compared (visually) the large-scale features of the atmospheric circulation during IOP16 between GDAS and COSMO-CLM. Figure 2 shows IWV simulated by GDAS on the 26-Oct-2012 at 0000 UTC and 1200 UTC, analogously to Fig.4 the manuscript without the wind fields. At 0000 UTC (Fig.2.a) IWV up to 40 mm is located south of France over the Gulf of Lions. At 1200 UTC IWV up to 40 mm is located between Sardinia and Sicily, similarly to the distribution showed by COSMO-CLM. Both time steps show similar IWV spatial distributions between GDAS and COSMO-CLM.

a)   b)

[Figure]

*Figure 2. IWV simulated by GDAS on the 26-Oct-2012 at 0000 UTC (a) and at 1200 UTC (b).*

Since the GDAS (0.5°) product is obtained using such large number and types of observations, and given the large similarities between the spatial distributions of IWV between COSMO-CLM we are confident of the GDAS accuracy providing the large-scale conditions of pressure distribution and mean flow, which are the elements used to explain the moisture transport from the north African coast.

3-TECHNICAL CORRECTIONS

We have accepted all corrections and comments in the revised version of the manuscript. Where needed, a short clarification is added here.

*L16, 88, 501, and other places: Hours and minutes are not appropriate frequency units.*

We have adapted the text to the term "temporal resolution" of the GPS nudging, instead of "frequency".

4-REFERENCES

Andrei, C., Chen, R. Assessment of time-series of troposphere zenith delays derived from the Global Data Assimilation System numerical weather model. GPS Solut 13, 109–117 (2009). https://doi.org/10.1007/s10291-008-0104-1

Dey, C. H., and L. Lhttps://doi.org/10.1175/WAF-D-16-0202.1.. Morone, 1985: Evolution of the National Meteorological Center Global Data Assimilation System: January 1982–December 1983. *Mon. Wea. Rev.*, **113**, 304–318, https://doi.org/10.1175/1520-0493(1985)113<0304:EOTNMC>2.0.CO;2.

Schraff, C. and Hess, R.: A Description of the Nonhydrostatic Regional COSMO-Model Part III: Data Assimilation, DeutscherWetterdienst, P.O. Box 100465, 63004 Offenbach, Germany, 2012.

Hoover, B. T., D. A. Santek, A. Daloz, Y. Zhong, R. Dworak, R. A. Petersen, and A. Collard, 2017: Forecast Impact of Assimilating Aircraft WVSS-II Water Vapor Mixing Ratio Observations in the Global Data Assimilation System (GDAS). *Wea. Forecasting*, **32**, 1603–1611,

Kleist, D. T., D. F. Parrish, J. C. Derber, R. Treadon, W. Wu, and S. Lord, 2009: Introduction of the GSI into the NCEP Global Data Assimilation System. *Wea. Forecasting*, **24**, 1691–1705, https://doi.org/10.1175/2009WAF2222201.1.

Liu, L., K. Garrett, E. S. Maddy, and S. Boukabara, 2018: Impact Assessment of Assimilating NASA's RapidScat Surface Wind Retrievals in the NOAA Global Data Assimilation System. *Mon. Wea. Rev.*, **146**, 929–942, https://doi.org/10.1175/MWR-D-16-0179.1.

Lukens, E. L., Ide. K., Garrett, K. and Wang, L., 2019: Assessment of Stratospheric Balloon Observations Towards Assimilation in NOAA's GSI-Based Global Data Assimilation System. AGU 2019

**Answers to reviewer 2 (Second revision)**

We would like to thank the anonymous reviewer for the corrections, mostly technical included, in the last revision report. We have accepted all comments and adapted the manuscript accordingly. The changes can be seen in the "track_changes" document.

Best regards.

[revised manuscript text omitted]

---

## Author Response (AR3)

**Answer to editor**

Dear Ms. Ducrocq

Thank you very much for your last comments and technical corrections. We have accepted all of them and the new version has been uploaded to the NHESS platform.

We are very happy of your (and reviewers) acceptance of the paper and we are looking forward to continue working with NHESS in a near future.

Sincerely yours,

The authors